# Affinity-tagged SMAD1 and SMAD5 mouse lines reveal transcriptional reprogramming mechanisms during early pregnancy

Zian Liao[1,2,3,4], Suni Tang[1,3], Kaori Nozawa[1,4], Keisuke Shimada[5], Masahito Ikawa[5], Diana Monsivais[1,4]*, Martin Matzuk[1,2,3,4]*

[1]Department of Pathology & Immunology, Baylor College of Medicine, Houston, United States; [2]Graduate Program of Genetics and Genomics, Baylor College of Medicine, Houston, United States; [3]Department of Molecular and Human Genetics, Baylor College of Medicine, Houston, United States; [4]Center for Drug Discovery, Baylor College of Medicine, Houston, United States; [5]Research Institute for Microbial Diseases, Osaka University, Osaka, Japan

*For correspondence:
diana.monsivais@bcm.edu (DM);
mmatzuk@bcm.edu (MM)

Competing interest: The authors declare that no competing interests exist.

**Abstract** Endometrial decidualization, a prerequisite for successful pregnancies, relies on transcriptional reprogramming driven by progesterone receptor (PR) and bone morphogenetic protein (BMP)-SMAD1/SMAD5 signaling pathways. Despite their critical roles in early pregnancy, how these pathways intersect in reprogramming the endometrium into a receptive state remains unclear. To define how SMAD1 and/or SMAD5 integrate BMP signaling in the uterus during early pregnancy, we generated two novel transgenic mouse lines with affinity tags inserted into the endogenous SMAD1 and SMAD5 loci (*Smad1*[HA/HA] and *Smad5*[PA/PA]). By profiling the genome-wide distribution of SMAD1, SMAD5, and PR in the mouse uterus, we demonstrated the unique and shared roles of SMAD1 and SMAD5 during the window of implantation. We also showed the presence of a conserved SMAD1, SMAD5, and PR genomic binding signature in the uterus during early pregnancy. To functionally characterize the translational aspects of our findings, we demonstrated that SMAD1/5 knockdown in human endometrial stromal cells suppressed expressions of canonical decidual markers (*IGFBP1, PRL, FOXO1*) and PR-responsive genes (*RORB, KLF15*). Here, our studies provide novel tools to study BMP signaling pathways and highlight the fundamental roles of SMAD1/5 in mediating both BMP signaling pathways and the transcriptional response to progesterone (P4) during early pregnancy.

## eLife assessment

This study presents two **valuable** new mouse models that individually tag proteins from the SMAD family to identify distinct roles during early pregnancy. **Convincing** evidence is provided that SMAD1 and SMAD5 target many of the same genomic regions as each other and the progesterone receptor. Given the broad effect of these signaling pathways in multiple systems, these new tools will most likely interest readers across biological disciplines.

## Introduction

Infertility is an emerging health issue that affects approximately 15% of couples (*Boivin et al., 2007*). One in five women aged 15–49 years old with no prior births suffers from infertility in the United

States (*Martinez et al., 2018*). One important factor affecting fertility is failed embryo implantation or subsequent post-implantation loss due to endometrial defects. This is evident from the high number of failed pregnancies, with as many as 15% of pregnancies resulting in early pregnancy losses (*Wang et al., 2003*). Understanding the molecular mechanism of how the maternal endometrium becomes suitable for embryo implantation and eventual decidualization will be the key to eradicating global concerns related to infertility and early pregnancy losses.

The transforming growth factor β (TGFβ) family plays diverse roles in development, physiology, and pathophysiology (*Chang et al., 2002*; *Monsivais et al., 2017b*), and in particular, signaling pathways downstream of the bone morphogenetic protein (BMP) subfamily are essential for decidual formation (*Lee et al., 2007*; *Monsivais et al., 2017a*). There are more than 30 TGFβ family ligands, and these ligands signal through complexes of transmembrane type 1 activin-like kinase (ALK) receptors (ALK1, ALK2, ALK3, ALK6) and transmembrane type 2 receptors (BMPR2, ACVR2A, ACVR2B) and then phosphorylate downstream SMAD1 and SMAD5 proteins. Phosphorylated SMAD1/5 form heteromeric complexes with SMAD4 and translocate to the nucleus to induce specific transcriptional programs. Our laboratory and others have used genetically engineered mouse models with deletions of ligands, receptors, and downstream effectors of BMP signaling pathways to establish that BMP signaling pathways are major regulators of early pregnancy (*Lee et al., 2007*; *Monsivais et al., 2017a*; *Nagashima et al., 2013*; *Monsivais et al., 2021*; *Matzuk et al., 1995*; *Clementi et al., 2013*; *Monsivais et al., 2016*).

A successful pregnancy begins with reciprocal crosstalk between the maternal endometrium and the new blastocyst during the peri-implantation window. Effective implantation requires precise synchronization between the development of the blastocyst and the transformation of the maternal endometrium into a functional decidua. Endometrial stromal fibroblasts undergo the decidualization process in which they differentiate into unique secretory decidual cells that offer a supportive and immune-privileged microenvironment required for embryo implantation and placental development. Decidualizing stromal cells can react to individual embryos in a way that either supports the implantation and subsequent embryonic development or exerts early rejection (*Teklenburg et al., 2010*; *Brosens et al., 2014*). Aberrant decidualization processes are observed in patients with recurrent pregnancy loss (RPL), displaying a disordered pro-inflammatory response, decreased induction of decidual marker genes, and abnormal responses to embryonic human chorionic gonadotropin (*Teklenburg et al., 2010*; *Weimar et al., 2012*; *Salker et al., 2011*). In addition to affecting early pregnancy outcomes, defective decidualization is also involved in the maternal etiology of preeclampsia, causing an abnormal placental phenotype (*Garrido-Gomez et al., 2017*; *Garrido-Gomez et al., 2020*). The process of decidualization is tightly regulated by hormone signaling pathways (estrogen, E2, and progesterone, P4), as well as by BMP signaling pathways. Our recent studies found that endometrial *Smad1* deletion had no significant effect on fertility, *Smad5* conditional deletion resulted in subfertility, while double *Smad1/5* conditional deletion led to infertility due to implantation and decidualization defects (*Monsivais et al., 2021*). The uteri of mice with double conditional *Smad1/5* deletion also displayed decreased response to P4 during the window of implantation, suggesting synergy between the two pathways. However, the mechanistic genomic actions of SMAD1 and/or SMAD5 in the uterus have not been explored, partly because there are no specific antibodies that distinguish phospho-SMAD1 versus phospho-SMAD5.

In this study, we define how SMAD1/5 instructs the decidualization process using genomic approaches in newly generated transgenic mouse lines. We inspect the potential crosstalk between P4 and BMP signaling pathways mediated by SMAD1/5. Together, our study demonstrates that SMAD1 and SMAD5 exhibit shared and unique genomic binding features and further reveals that SMAD1/5 contributes to the P4 response through transcriptional reprogramming during decidualization.

## Results

### Generation of mouse models with global HA-tagged SMAD1 and PA-tagged SMAD5 proteins

Activation of BMP signaling pathways has been established as one of the hallmarks of the decidualization process (*Gellersen and Brosens, 2014*; *Magro-Lopez and Muñoz-Fernández, 2021*). Canonically, SMAD1/5 are regarded as downstream effectors of BMP2 signaling pathways to regulate

decidual-specific gene expressions (*Lee et al., 2007*; *Li et al., 2007*). However, our recent findings demonstrated that SMAD1/5 can also affect the sensitivity of the endometrium toward E2 and P4 stimulation (*Monsivais et al., 2021*). Since we observed phenotypical differences between uterine-specific single SMAD1 and single SMAD5 deletion mice, it is beneficial to delineate the role of SMAD1 and SMAD5 in mediating P4 responses during early pregnancy. We used CRISPR technology to generate genetically engineered knock-in mice with an HA-tagged *Smad1* allele (herein called *Smad1*$^{HA/HA}$) and PA-tagged *Smad5* allele (herein called *Smad5*$^{PA/PA}$) as shown in *Figure 1A and B*. The HA tag and the PA tag were inserted into the N-terminus of the SMAD1 and SMAD5 proteins, respectively. Sanger sequencing was used to confirm genomic insertion (*Figure 1A and B*, *Figure 1—figure supplement 1*). To validate the global detection of tagged proteins, we performed immunoprecipitation followed by western blot analysis on different tissues from *Smad1*$^{HA/HA}$ and *Smad5*$^{PA/PA}$ mice. We confirmed the HA and PA antibodies can readily detect HA-tagged SMAD1 and PA-tagged SMAD5 proteins at the predicted size (*Figure 1C and D*). We also demonstrated the molecular size and expression pattern of HA antibody-detected SMAD1 protein was comparable to the SMAD1 antibody-detected SMAD1 protein across different tissue types. Similarly, PA antibody showed comparable signal intensity to the SMAD5 antibody in detecting SMAD5 protein across different tissue types (*Figure 1C and D*). Thus, we successfully generated viable mouse models with global HA-tagged SMAD1 and PA-tagged SMAD5 proteins.

## SMAD1 and SMAD5 exhibit shared and unique genomic binding sites during decidualization

The BMP signaling pathway regulates multiple key events during early pregnancy (*Monsivais et al., 2017b*), mediated through receptor-regulated SMAD proteins, including SMAD1 and SMAD5. As transducers of the BMP signaling pathway, phosphorylated SMAD1 and SMAD5 form homomeric complexes and then couple with SMAD4 to assemble hetero-oligomeric complexes in the nucleus to execute transcription programs. Our previous studies revealed that conditional ablation of SMAD1 and SMAD5 in the uterus decreased P4 response during the peri-implantation period, suggesting that the transcriptional activities of PR depend on BMP/SMAD1/5 signaling (*Monsivais et al., 2021*). Furthermore, previous genome-wide PR binding studies show that SMAD1 and SMAD4 binding motifs are enriched in PR binding sites in the uterus (*Rubel et al., 2012*).

To determine the shared and unique transcriptional regulomes of SMAD1 and SMAD5 contributing to the diverse effects of BMP and P4 signaling pathways during decidualization, we first utilized Cleavage Under Targets & Release Using Nuclease (CUT&RUN) (*Skene and Henikoff, 2017*) coupled with next-generation sequencing to profile genomic loci bound by SMAD1, SMAD5, and PR from mouse uterine tissues. We performed CUT&RUN on the uterine tissues collected at 4.5 days post coitus (dpc), the time when the fertilized embryo reaches the uterus physically and initiates the decidualization program (*Ramathal et al., 2010*; *Figure 2A*). After aligning CUT&RUN reads to the mm10 mouse genome, we called peaks using Sparse Enrichment Analysis for CUT&RUN (SEACR) (*Meers et al., 2019*). To identify high-confidence peaks, background noise was normalized to IgG and the stringent criteria for peak calling in SEACR were used. After merging common peaks from two biological replicates, we identified 118,778 peaks for SMAD1 and 166,025 total peaks for SMAD5. We visualized the enrichment of SMAD1 and SMAD5 peaks to the overall aligned chromatin regions with clustering for preferential enrichment, as shown in *Figure 2B*. Peaks in cluster 1 exhibit a shared enrichment for both SMAD1 and SMAD5, whereas clusters 2 and 3 demonstrate preferential enrichment for SMAD5 and SMAD1, respectively. We found that 7.55% of SMAD1 peaks and 9.53% of SMAD5 peaks were located within the ±3 kb of the promoter regions (*Figure 2C and D*). This corresponded to 10,368 genes that were directly bound by SMAD1 at the promoter regions (±3 kb), whereas 18,270 genes were directly bound by SMAD5 at the promoter regions (±3 kb). Among these, 4933 genes were found in common between SMAD1 (47.5%) and SMAD5 (27.0%), while 2744 and 7427 genes were found to be uniquely bound by SMAD1 and SMAD5, respectively, providing evidence for the shared and unique functions of SMAD1 and SMAD5 at the transcriptional level (*Figure 2—figure supplement 1*). Hence, interpreting how the binding events correlate to biological activity requires comparisons with gene expression profiling in a tissue-specific manner.

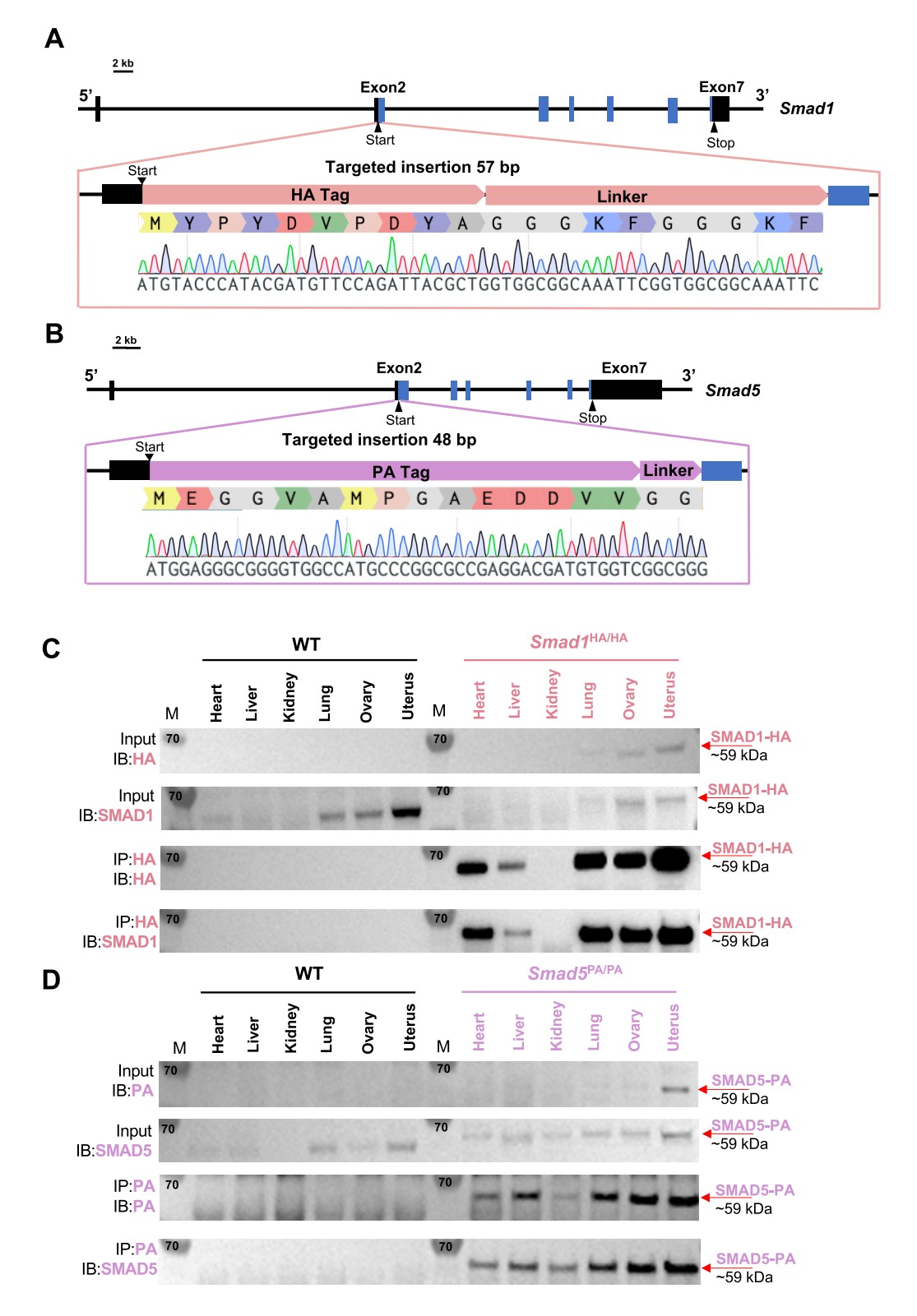

**Figure 1.** Mouse models with global HA-tagged SMAD1 and PA-tagged SMAD5 proteins. (**A, B**) Schematic approaches for generating Smad1[HA/HA] and Smad5[PA/PA] knock-in mouse lines. Sanger sequencing of the genotyping results is included as validation of knock-in sequence. Black and blue boxes indicate untranslated and coding regions, respectively. (**C, D**) Immunoblot (IB) analysis of the immunoprecipitation (IP) of HA-tagged SMAD1 and PA-

*Figure 1 continued on next page*

Figure 1 continued

tagged SMAD5 proteins from different tissues of the tagged mouse lines. Wild-type (WT) mice were used as negative controls. Antibodies used for IB and IP are as labeled. Targeted bands of SMAD1 and SMAD5 are indicated by red arrows.

The online version of this article includes the following source data and figure supplement(s) for figure 1:

Source data 1. Raw uncropped western blot images for *Figure 1C and D*, without labels.

Source data 2. Uncropped western blot images for *Figure 1C*, with labels.

Source data 3. Uncropped western blot images for *Figure 1D*, with labels.

Figure supplement 1. Genotype of the knock-in mouse lines.

## Identification of direct target genes of SMAD1 and SMAD5 during early pregnancy

To pinpoint the direct target genes of SMAD1 and SMAD5, we integrated transcriptomic data from previously published (*Monsivais et al., 2021*) SMAD1/5 double conditional knockout mice using progesterone receptor cre (SMAD1/5 cKO) (GSE152675) with SMAD1 and SMAD5 genomic data from this article. We cross-compared the differentially expressed genes in the transcriptomic data to the SMAD1 and SMAD5 bound genes, respectively. Among the 805 significantly upregulated genes, we identified 449 genes that were both significantly upregulated upon SMAD1/5 depletion and were directly bound by SMAD1 and SMAD5, whereas 187 of the upregulated genes were bound by SMAD5 only and 30 were bound only by SMAD1 (*Figure 3A*). Among the 683 significantly downregulated genes, we identified 523 genes that were both significantly downregulated upon SMAD1/5 depletion and were directly bound by SMAD1 and SMAD5, whereas 83 of the downregulated genes were bound by SMAD5 only and 13 were bound by SMAD1 only (*Figure 3B*, *Supplementary file 3b*).

Next, we utilized Binding and Expression Target Analysis (BETA) algorithm (*Wang et al., 2013*) to perform motif enrichment analysis of the direct target genes to identify putative co-factors working together with SMAD1 and SMAD5 in controlling gene expression (*Figure 3C and D*, *Supplementary files 1* and *2*). 'Up-targets' represent genes that were upregulated in the SMAD1/5 cKO mouse uteri and showed either a SMAD1 or a SMAD5 binding site in the genomic profiling data. Similarly, 'down-targets' represent genes that were downregulated in the SMAD1/5 cKO mouse uteri and displayed either a SMAD1 or a SMAD5 binding site. Thus, motifs enriched in the 'up-targets' indicate potential repressive SMAD1/5 co-factors while motifs enriched in the 'down-targets' indicate potential SMAD1/5 co-activators. Among the 'up-targets' of SMAD1, MYB Proto-Oncogene (*Myb*)/ MYB Proto-Oncogene Like 1(*Mybl1*) motif was the most highly enriched with a p-value of 1.85E-02. *Myb* and *Mybl1* transcription factors belong to MYB gene family, which has been well-defined in controlling cell survival, proliferation, and differentiation in cancer (*Cicirò and Sala, 2021*). In addition, they have also been reported to be E2 induced in human uterine leiomyoma samples (*Swartz et al., 2005*). Homeobox containing 1 (*Hmbox1*) and Krüppel-like factor (*Klf*) family members (*Klf4/Klf1/ Klf12*) were also identified as potential repressive co-factors of SMAD1 with p-values of 2.85E-02 and 3.75E-02, respectively (*Figure 3C*). Of note, *KLF4* has been reported to inhibit the binding activity of estrogen receptor α (ERα) to estrogen response elements in promoter regions (*Akaogi et al., 2009*). Among the 'up-targets' of SMAD5, EBF Transcription Factor 1 (*Ebf1*) motif was the most enriched with a p-value of 1.57E-02. Interestingly, *Ebf1* can directly repress the transcription of Forkhead box protein O1 (*Foxo1*) (*Timblin and Schlissel, 2013*). It is also recognized as downstream effector of steroid hormone receptors in the mouse uterus (*Pan et al., 2006*). Additionally, motifs from transcription factors *Zfp128* and *Otx1* were also significantly enriched in the upregulated genes bound by SMAD5 (*Figure 3D*). Taken together, our enrichment analysis provided robust evidence for identifying novel co-factors of SMAD1/5, and such co-regulating mechanisms are in line with the unopposed E2 response observed in the SMAD1/5 cKO mice (*Monsivais et al., 2021*). Furthermore, odd-skipped-related genes (*Osr1* and *Osr2*) were identified as potential co-activators for SMAD1. *Osr2* has been reported to be highly expressed in the human endometrium (*Fagerberg et al., 2014*), and it was also abundantly detected at the protein level in the human decidual tissues (*Ma et al., 2022*). Decreased OSR2 level was observed in the patients with recurrent spontaneous abortion and knockdown of *OSR2* impairs the decidualization process in the human endometrial stromal cells (EnSCs) (*Ma et al., 2022*). Moreover, OSR1 has been reported to suppress BMP4 expression, which in turn reduced the Wnt/β-catenin signaling pathways during lung development in *Xenopus* (*Rankin et al., 2012*). Apart

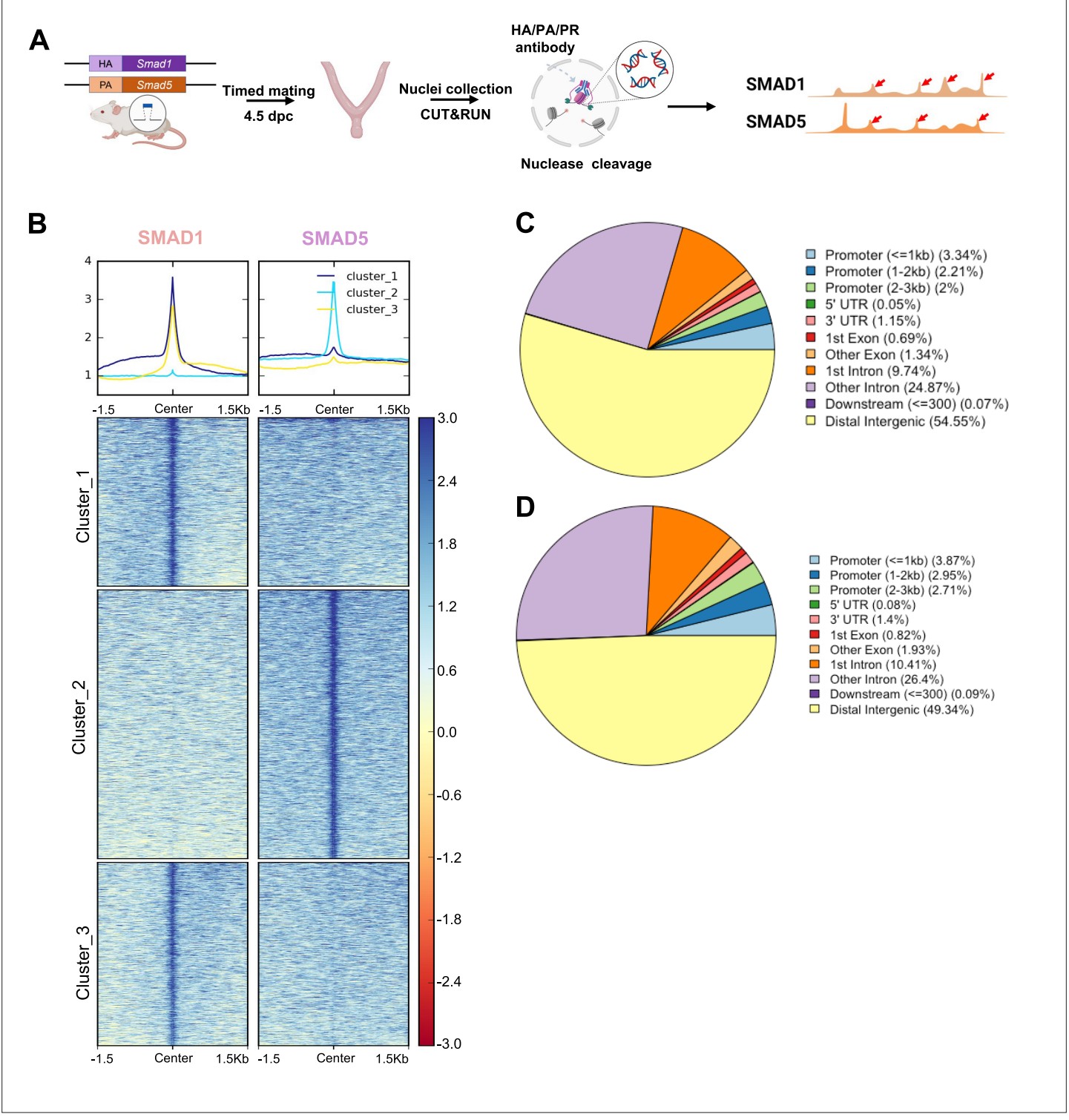

**Figure 2.** Genomic profiling of SMAD1 and SMAD5 binding sites during decidualization in vivo. (**A**) Diagram outlining experimental approaches for tissue collection, processing, and CUT&RUN. (**B**) Heatmaps and summary plots showing the enrichment of SMAD1 and SMAD5 binding peaks from one exemplary replicate. Clustering was conducted using k-means algorithm. The colors in the summary plots correspond to clusters labeled in the heatmap below. (**C, D**) Feature distribution of the annotated peaks for the SMAD1 (**C**) binding sites and SMAD5 (**D**) binding sites.

The online version of this article includes the following figure supplement(s) for figure 2:

**Figure supplement 1.** Shared and unique genes bound by SMAD1 or SMAD5 in the promoter region.

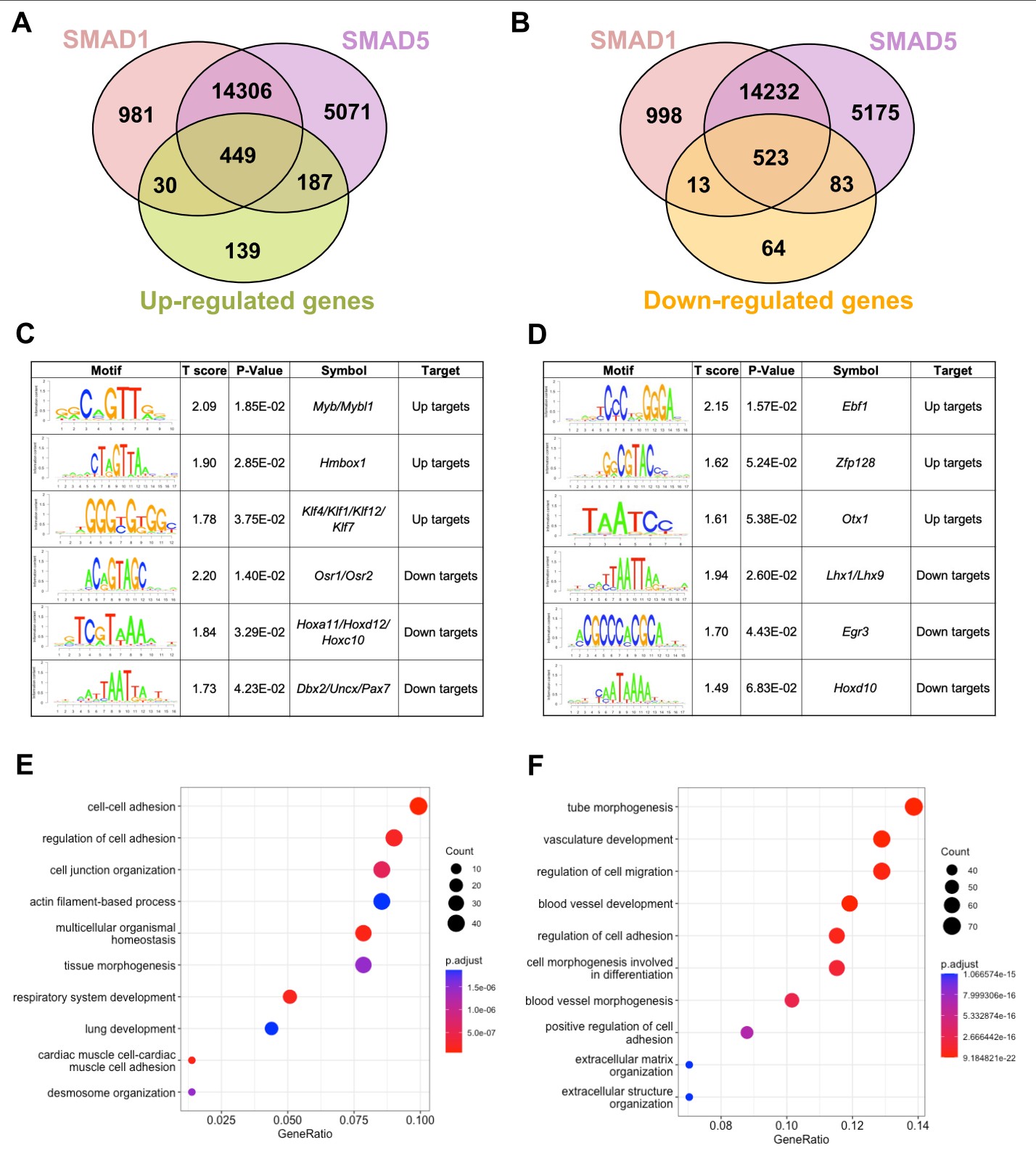

**Figure 3.** SMAD1 and SMAD5 show unique direct target genes during early pregnancy. (**A, B**) Venn diagrams showing the shared and unique direct up-target genes (**A**) and down-target genes (**B**) of SMAD1, SMAD5. Numbers indicate genes numbers. (**C, D**) Motif enrichment analysis from the up-targets and down-targets for SMAD1 (**C**) and SMAD5 (**D**). (**E, F**) Dot plot showing Gene Ontology enrichment analysis of shared direct target genes of SMAD1/5 from the up-targets (**E**) and the down-targets (**F**), respectively. Dot size represents the gene ratio in the enriched categories compared to background genes, and dot colors reflect p-value.

from Osr family, motifs in the Homeobox genes (HOX) were found to be enriched in the 'down-targets' from both SMAD1 and SMAD5 datasets. Specifically, *Hoxa11/Hoxd12/Hoxc10* were predicted to be co-activators for SMAD1 while *Hoxd10* was indicated to be closely interacting with SMAD5. Indeed, HOX genes are critical for endometrial development in normal and disease conditions and are essential during the establishment of pregnancy (*Du and Taylor, 2015*; *Cakmak and Taylor, 2010*; *Taylor et al., 1999*; *He et al., 2018*).

With direct target genes of SMAD1 and SMAD5 identified, we then analyzed the Gene Ontology enrichment for the SMAD1/5 shared up-targets and down-targets, respectively. We found that 'up-target' genes exhibit enrichment for regulation of cell–cell adhesion, cell junction organization, and desmosome organization (*Figure 3E*, *Supplementary file 3c*). Moreover, among the 'down-target' genes, we found the enrichment for blood vessel/vasculature development and extracellular matrix organization categories (*Figure 3F*, *Supplementary file 3d*). Indeed, during early pregnancy, the stimulation from corpus-luteum-derived P4 enabled the endometrium to be transformed to a receptive state, which allows subsequent embryo attachment and develop through the epithelium into the stromal sections (*Gellersen and Brosens, 2014*). During this process, apportioned direct cell–cell contacts are ensured by tight and adherent junctions and such interactions are key in facilitating implantation and embryo invasion. In accordance with our findings, desmosomes and adherens junctions were extensively described to decline in the early pregnancy period, which facilitates the invasion of trophoblast through the epithelial layer (*Illingworth et al., 2000*; *Paria et al., 1999*; *Potter et al., 1996*; *Grund and Grümmer, 2018*). In addition, the stromal compartment of the endometrium also undergoes profound vascular remodeling. Precise regulations of angiogenesis are required to establish an extensive vascular network, which is essential to ensure blood supply and successful embryonic development (*Schatz et al., 2016*; *Evans et al., 2016*). Collectively, our findings present evidence that emphasizes the shared roles of SMAD1 and SMAD5 in facilitating endometrial transitions during early pregnancy.

## Direct target genes of SMAD1 and SMAD5 maintain the homeostasis of uterine function

To discover novel direct target genes of SMAD1/5, we visualized key genes of interest from the up-targets and down-targets. As shown in *Figure 4A*, data from RNA-seq represents the decrease of several 'down-targets' in the SMAD1/5 cKO mouse uteri, including retinoic acid-related orphan receptor B (*Rorb*), follistatin (*Fst*), lymphoid enhancer binding factor 1 (*Lef1*), and insulin-like growth factor 1 (*Igf1*). Integrative Genomics Viewer (IGV) track view shows the exemplary SMAD1/5 binding activities near the promoter regions of *Rorb* and *Fst* (*Figure 4B*), demonstrating that these genes are bona fide direct target genes of SMAD1/5. *Rorb* belongs to the nuclear receptor families in the retinoic acid (RA) signaling pathways (*Stehlin-Gaon et al., 2003*) and is considered a marker for mesenchymal progenitor cells in the stroma compartment of the endometrium (*Spitzer et al., 2012*). In murine models, deficient RA signaling through the perturbation of RA receptor in the uterus leads to implantation and decidualization failure (*Yin et al., 2021*). *Fst* binds several TGFβ family ligands and thereby inhibits TGFβ family signaling extracellularly (*Chang, 2016*). Under physiological conditions, *Fst* is upregulated in the decidua during early pregnancy. Conditional deletion of *Fst* in the mouse uterus results in severe subfertility with a phenotype of non-receptive epithelium and poorly differentiated stroma (*Fullerton et al., 2017*). Notably, RA signaling deficiency also decreases *Fst* levels in the uterus and systematic administration of FST can fully rescue the deficient-decidualization phenotype but not the non-receptive phenotype observed in the RA receptor mutant mice (*Yin et al., 2021*). Our results suggest a direct relationship between BMP and RA signaling pathway, accomplished by SMAD1/5 at the transcriptional level, likely establishing a positive signaling feedback loop. Apart from being a crucial transcriptional activator, SMAD1/5 also plays a role in repressing key gene expression pathways. Shown in *Figure 4C*, upon the deletion of SMAD1/5 in the mouse uteri, several E2-responsive genes were significantly upregulated, including fibroblast growth factor receptor 2 (*Fgfr2*), matrix metallopeptidase 7 (*Mmp7*), and Wnt family member 7B (*Wnt7b*). In addition, *Inhbb*, a downstream target of *Fst* (*Fullerton et al., 2017*), is also a target gene of SMAD1/5 that resulted in transcriptional repression. SMAD1/5 binding on the *Fgfr2* and *Mmp7* genes are exemplified in an IGV track view in *Figure 4D*. *Fgfr2* and its ligands regulate epithelial cell proliferation and differentiation. Components of the fibroblast growth factor (*Fgf*) signaling pathway are cyclically expressed in the uterus and act as

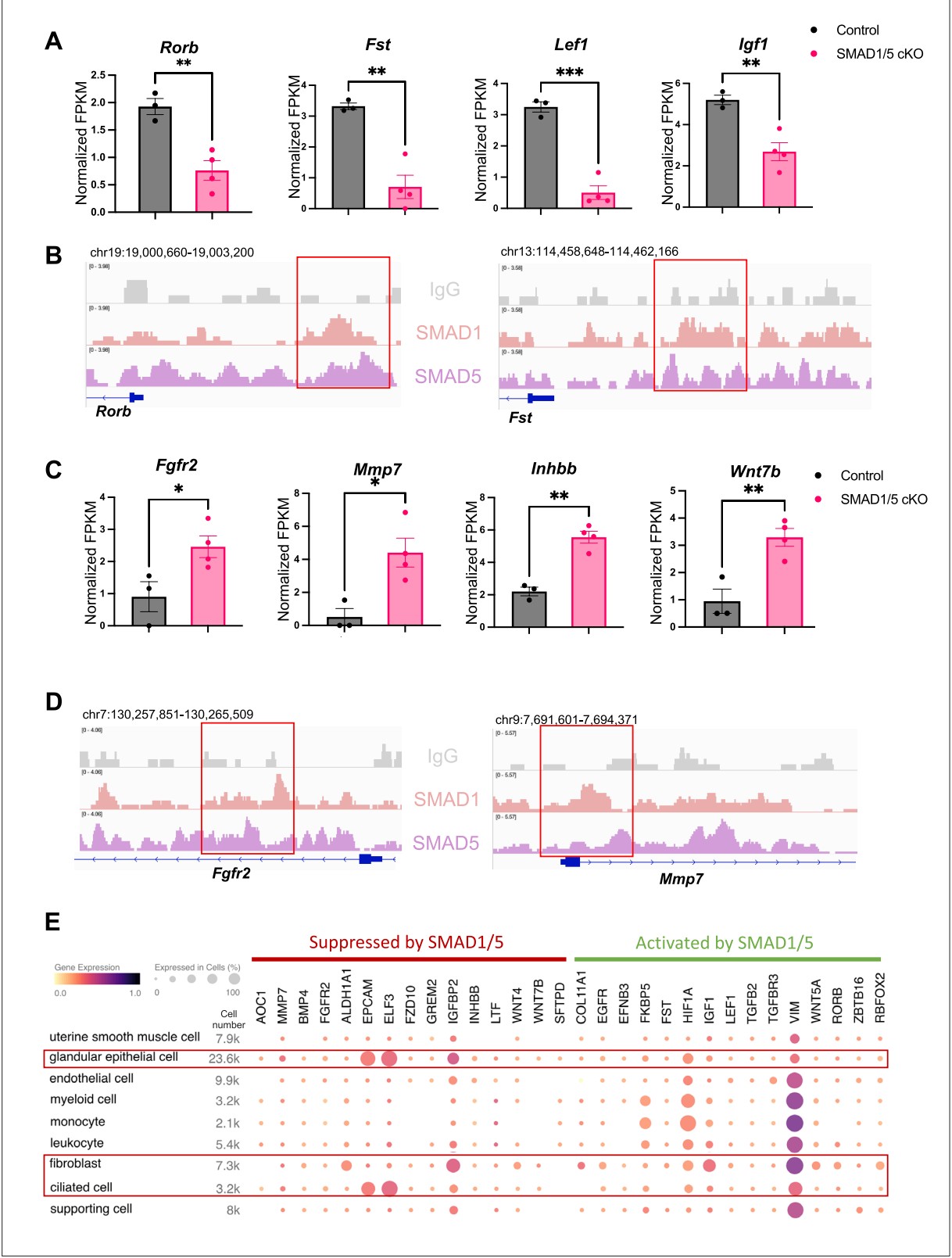

**Figure 4.** Direct target genes of SMAD1/5 mediate uterine homeostasis. (**A**) Bar graph of normalized fragments per kilobase of transcript per million mapped reads (FPKM) of downregulated transcripts in the control and SMAD1/5 conditional knockout (cKO) groups as indicated by the label. Histograms represent average ± SEM of experiments uteri from control mice (N = 3) and SMAD1/5 cKO mice (N = 4). Analyzed by an unpaired *t*-test. (**B**) Integrative Genomics Viewer (IGV) track view of SMAD1, SMAD5 binding activities. Gene loci are as indicated in the figure, genomic coordinates are

*Figure 4 continued on next page*

Figure 4 continued

annotated in mm10. (**C**) Bar graph of FPKM of upregulated transcripts in the control and SMAD1/5 cKO groups as indicated by the label. (**D**) IGV track view of SMAD1, SMAD5 binding activities. Gene loci are as indicated in the figure, genomic coordinates are annotated in mm10. (**E**) Dot plot showing the gene expression pattern of the key SMAD1/5 direct target genes in different cell types from published human endometrium single-cell RNA-seq dataset.

The online version of this article includes the following figure supplement(s) for figure 4:

**Figure supplement 1.** Cell type compositions in the control and SMAD1/5 PR-Cre mice.

paracrine and/or autocrine mediators of epithelial–stromal interactions (*Li et al., 2011*; *Filant et al., 2014*). During early pregnancy in mice, P4 inhibits expression of *Fgf2* in the stromal cells, which is critical to counteract the E2-driven epithelial proliferation (*Li et al., 2011*). Similar observations are reported in gilts, where the expression of *Fgfr2* decreased alongside with increased parity of the sows (*Lim et al., 2017*). It is also noteworthy that loss of function of *Fgfr2* in the mouse uterus leads to luminal epithelial stratification and peri-implantation pregnancy loss (*Filant et al., 2014*). Moreover, *Mmp7* and *Wnt7b* are upregulated upon E2 stimulation and participate in the re-epithelialization of the endometrium and implantation process, respectively (*Russo et al., 2009*; *Tenvergert et al., 1992*; *Hayashi et al., 2009*). In accordance with the phenotype of hyperproliferative endometrial epithelium during early pregnancy, observed in SMAD1/5 cKO mice, we demonstrated that the suppression of key E2-responsive genes, such as *Fgfr2* and *Mmp7*, by SMAD1/5 maintains the precise balance between E2 and P4.

To explore the major cell types regulated by SMAD1/5, first, we used CIBERSORTx (*Newman et al., 2019*) to analyze and depict changes in the cell populations upon SMAD1/5 depletion in the mouse uterus during early pregnancy. By imputing the bulk uterine gene expression profiles to previously published mouse uterine single-cell datasets (*Yang et al., 2023*) using CIBERSORTx (*Newman et al., 2019*), we were able to compare changes across both samples and cell types upon the SMAD1/5 perturbation in the mouse uterus. We highlight the proportional increase in the epithelial cells, as well as the decrease in the decidual stromal cells and smooth muscle cells in mice lacking uterine SMAD1/5 during the peri-implantation phase (*Figure 4—figure supplement 1*). Such cell populational changes are in line with the phenotypical observations of decidualization failure and excessive proliferation in the epithelial compartment. In addition, to explore the expression patterns of SMAD1/5 direct targets in human, we profiled the expression levels of the key 'up-targets' and 'down-targets' in the different cell types of the human endometrium. Using previously published single-cell RNA-seq data of human endometrium (*Garcia-Alonso et al., 2021*), we visualized the expression patterns of suppressive targets and activating targets of SMAD1/5 (*Figure 4E*). Apart from the major epithelial and stromal compartments, SMAD1/5 target genes are also widely expressed in the immune cell populations. Such observations reinforced the importance of the BMP signaling pathways in establishing an immune-privileged environment at the maternal–fetal interface (*PrabhuDas et al., 2015*).

## SMAD1 and SMAD5 co-regulate PR target genes

SMAD1/5 cKO mice were infertile due to endometrial defects and displayed decreased P4 response during the peri-implantation period (*Monsivais et al., 2021*). Hence, we hypothesized that SMAD1 and SMAD5 act as co-regulators of P4-responsive genes during the window of implantation and are required for endometrial receptivity and decidualization. By determining the genomic co-occupancy of SMAD1, SMAD5, and PR, we aimed to clarify the transcriptional interplay between the BMP and P4 signaling pathways. To this end, we performed additional PR CUT&RUN experiments on the uteri of mice collected at 4.5 dpc and identified 134,737 peaks showing PR binding activities (*Figure 5A*). Based on the k-means clustering results of the peaks, we demonstrated clusters with shared occupancy between SMAD1/5 and PR (cluster 1), preferential deposition in the SMAD1 (cluster 2), SMAD5 (cluster 4), and PR (clusters 3 and 5), respectively. Interestingly, between clusters 3 and 5, although the primary enrichment is for PR, overall the signal intensities for SMAD5 are higher in cluster 5. Together with previous analysis on genes uniquely or commonly bound by SMAD1/5 (*Figure 2—figure supplement 1*), we speculate such observation can be attributed to a subset of the genes that are potentially co-regulated by SMAD5 and PR. From the gene perspective, we identified 7393 genes that were directly bound by PR at the promoter regions (±3 kb), among which 2596 genes were also

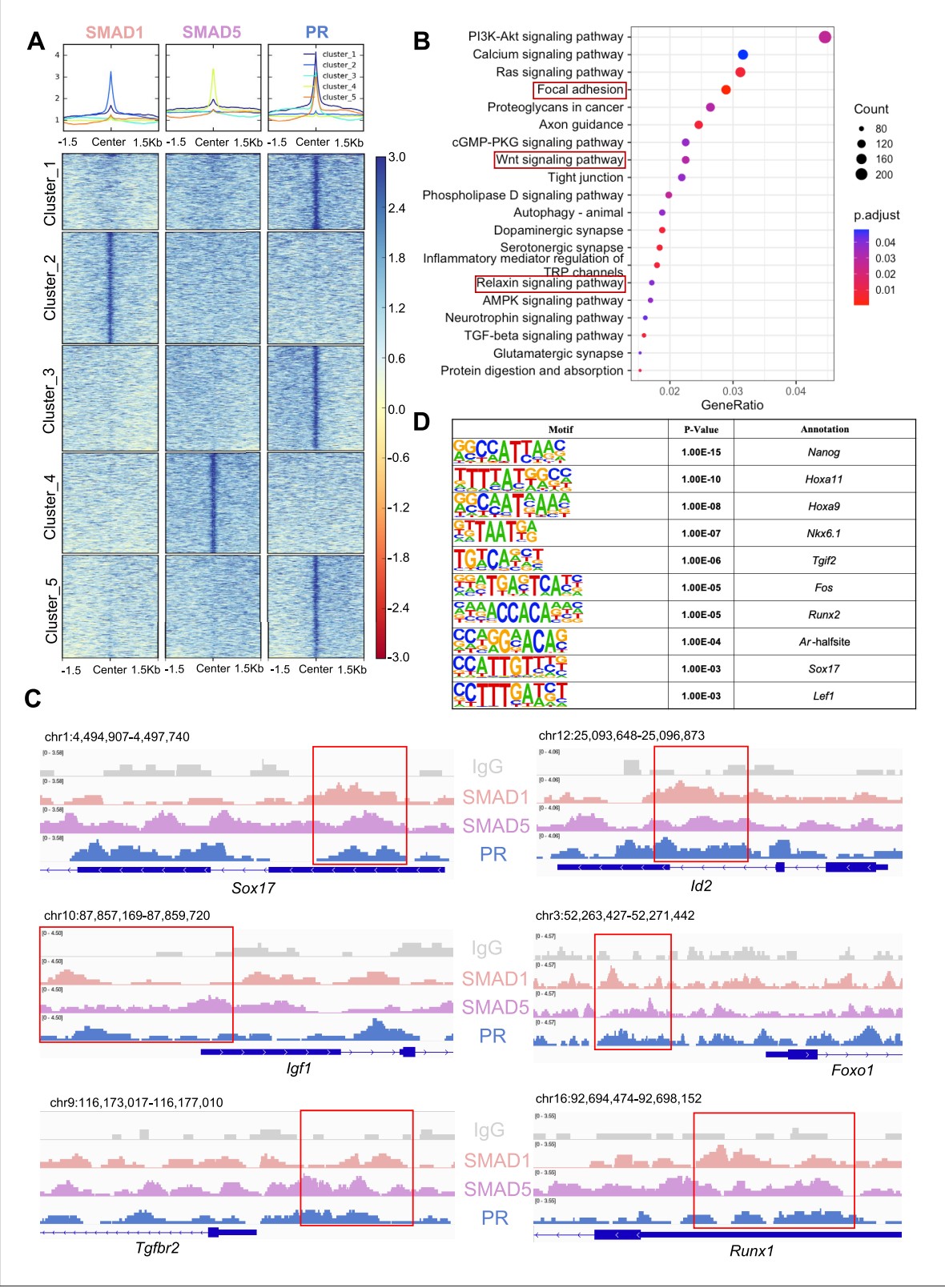

**Figure 5.** SMAD1 and SMAD5 co-regulate progesterone receptor (PR) target genes. (**A**) Heatmaps and summary plots showing the enrichment comparison between SMAD1, SMAD5, and PR binding peaks from one exemplary replicate. Clustering was conducted using k-means algorithm. The colors in the summary plots correspond to clusters labeled in the heatmap below. (**B**) Dot plot showing KEGG pathway enrichment analysis for shared genes bound by SMAD1, SMAD5, and PR. (**C**) Integrative Genomics Viewer (IGV) track view of SMAD1, SMAD5, and PR binding activities. Gene loci are

Figure 5 continued

as indicated in the figure, genomic coordinates are annotated in mm10. (**D**) Table of motif analysis results for shared peaks between SMAD1, SMAD5, and PR, with p-value and motif annotation specified for each motif.

The online version of this article includes the following figure supplement(s) for figure 5:

**Figure supplement 1.** Overlapping of SMAD1/5 with known transcription factors governing uterine homeostasis.

concurrently bound by both SMAD1 and SMAD5 at the promoter regions (±3 kb) (*Figure 5—figure supplement 1A*).

Next, we performed KEGG pathway enrichment for the genes co-bound by SMAD1, SMAD5, and PR. As expected, pathways critical for decidualization such as relaxin signaling pathways and WNT signaling cascade were identified in the enrichment results (*Figure 5B*, *Supplementary file 3e*). We visualized exemplary genes co-regulated by SMAD1, SMAD5, and PR and presented in the normalized IGV track view (*Figure 5C*). We demonstrated SMAD1, SMAD5, and PR showed co-occupancy at the loci of the SRY-box transcription factor 17 (*Sox17*), inhibitor of DNA binding 2 (*Id2*), forkhead box protein O1 (*Foxo1*), insulin-like growth factor 1 (*Igf1*), transforming growth factor beta receptor 2 (*Tgfbr2*), and RUNX family transcription factor 1 (*Runx1*) (*Figure 5C*). *Sox17* has been reported as one of the direct target genes of PR (*Rubel et al., 2012*) and is essential for uterine functions during implantation and early pregnancy (*Guimarães-Young et al., 2016*; *Hirate et al., 2016*). More recent studies also showed the importance of *Sox17* in regulating uterine epithelial–stromal crosstalk and its indispensable role in female fertility (*Wang et al., 2018*). We provided evidence that *Sox17* is also directly regulated by SMAD1/5 complexes. Our results indicated that *Id2*, considered as canonical direct transcriptional targets of BMP-SMAD signaling (*Hollnagel et al., 1999*; *Miyazono and Miyazawa, 2002*), is also regulated by PR. We also confirmed that known P4-responsive genes such as *Tgfbr2* (*Holloran et al., 2020*) and *Runx1* (*Dinh et al., 2023*), as well as decidual markers such as *Foxo1* (*Vasquez et al., 2018*) and *Igf1* (*Shi et al., 2022*), were co-regulated by SMAD1, SMAD5, and PR (*Figure 5C*).

To identify additional transcription factors that are associated with the regulatory interplay between SMAD1/5 and PR during decidualization, we performed unbiased motif analysis on the shared CUT&RUN peaks between SMAD1/5 and PR. We reported the top 10 transcription factors harboring the enriched motifs, including NANOG, Homeobox A protein family (HOXA11 and HOXA9), NK6 homeobox 1(NKX6.1), TGFB-induced factor homeobox 2 (TGIF2), FOS, RUNX family transcription factor 2 (RUNX2), androgen receptor (AR), SOX17, and lymphoid enhancer-binding factor 1 (LEF1) (*Figure 5D*, *Supplementary file 3f*). Many of these putative interactors have been reported to interact with the SMAD proteins in other biological processes. For example, NANOG interacts with SMAD1 during mesoderm differentiation (*Suzuki et al., 2006*). HOXA9 forms heterodimers with SMAD4, leading to BMP-driven initiation of transcription from the mouse *Opn* promoter in vitro (*Shi et al., 2001*; *Shi et al., 1999*). Transcription factor AP-1 family (FOS) and RUNX2, as well as β-catenin/Lef1 complex, increase the effectiveness and specificity of DNA binding activities of SMAD1/5 in response to BMP ligand stimuli (*Feng and Derynck, 2005*; *Massagué et al., 2005*; *Derynck and Budi, 2019*). To further evaluate the key roles of SMAD1/5 as major uterine transcription regulators, we cross-compared the genomic binding sites of SMAD1/5 with known key transcription factors, namely aforementioned SOX17 (*Figure 5—figure supplement 1B*, *Supplementary file 3g*), as well as NR2F2 (*Figure 5—figure supplement 1C*, *Supplementary file 3h*), an essential regulator of hormonal response (*Lee et al., 2010*), using our CUT&RUN data sets and published mouse uterine SOX17 and NR2F2 ChIP-seq data sets (GSE118328, GSE232583). Among the annotated genes, 5402 genes are shared between SMAD1/5 and SOX17, and 1922 genes are shared between SMAD1/5 and NR2F2. Such observations indicate a potential co-regulatory mechanism between SMAD1/5 and other key uterine transcription factors in maintaining appropriate uterine functions. Overall, our analyses demonstrate that the transcriptional activity of SMAD1, SMAD5, and PR coordinates the expression of key genes required for endometrial receptivity and decidualization.

## Decidualization of human endometrial stromal cells requires SMAD1/SMAD5

We next sought to functionally characterize the role of SMAD1/5 during decidualization in human EnSCs. To do so, we examined the effect of SMAD1/5 perturbations on the decidualization of primary human EnSCs. EnSCs were transfected with short interfering RNAs (siRNAs) targeting each gene (*SMAD1* and *SMAD5*) and subjected to in vitro decidualization by treatment with E2-cAMP-and MPA (EPC) for 4.5 d (*Figure 6A*, *Figure 6—figure supplement 1*). We hypothesized that the combined SMAD1/5 knockdown would impair the decidualization process significantly compared to cells treated with non-targeting siRNAs. Our results demonstrated that SMAD1/5 knockdown affected decidualization and led to significantly decreased expression of the canonical decidual markers, *PRL* and *IGFBP1* in EnSCs (*Figure 6B*). The PR co-regulator, *FOXO1* (*Vasquez et al., 2015*), also exhibited a significant decreasing trend in the siSMAD1/5 group. We also examined the expression level of the RA pathway regulator gene, *RORB*, and of the SMAD4-PR target gene, *KLF15* (*Monsivais et al., 2016*), following SMAD1/5 perturbation. We observed a significant decrease in both *RORB* and *KLF15* expression upon SMAD1/5 knockdown during in vitro decidualization treatment (*Figure 6C*). Taken together, our findings indicate that SMAD1/5 can modulate PR activity during decidualization and that this transcriptional cooperation is required for the in vitro decidualization of primary human EnSCs.

## Discussion

SMAD proteins are canonical transcription factors that are activated in response to TGFβ family signaling and mediate the biological effects of these pathophysiologically critical ligands (*Massagué et al., 2005*). While SMAD2 and SMAD3 are downstream of TGFβs, activins, and multiple other family ligands, SMAD1 and SMAD5 preferentially transduce BMP signaling pathways and are regarded as pivotal activators for many physiological processes, including bone development, cardiac conduction system development, and embryonic pattern specification (*Wu et al., 2016*; *van Weerd and Christoffels, 2016*; *Whitman, 1998*). Importantly, SMAD1 and SMAD5 are implicated in diverse female reproductive physiology and pathophysiology processes (*Monsivais et al., 2017b*; *Monsivais et al., 2021*; *Middlebrook et al., 2009*; *Pangas et al., 2008*; *Rodriguez et al., 2016*).

Due to high structural similarity, SMAD1/5 have been suggested to be redundant from the studies in ovarian biology and chondrogenesis (*Pangas et al., 2008*; *Retting et al., 2009*). However, other studies clearly demonstrated that SMAD1/5 have different roles in governing hematopoiesis and uterine functions (*Monsivais et al., 2021*; *McReynolds et al., 2007*). The DNA binding activities of SMAD1 and SMAD5 have not been readily distinguished from each other due to anti-phospho antibody limitations. To robustly define the roles of SMAD1/5 in regulating transcriptional programs in vivo, we produced two genetically engineered mouse models with global knock-in of an HA tag and a PA tag in the *Smad1* and *Smad5* loci, respectively. We showed that SMAD1 and SMAD5 not only have shared transcriptional activities but also have unique roles in uterine physiology. In agreement with previous studies showing that SMAD1/5 function is partially redundant (*Pangas et al., 2008*; *Retting et al., 2009*), we confirmed that SMAD1/5 share a total of 972 direct target genes in the uterus. Furthermore, we demonstrated that 43 genes were uniquely regulated by SMAD1 whereas 270 genes are specifically regulated by SMAD5 only. Our motif analysis also revealed distinct potential co-factors between SMAD1 and SMAD5, providing evidence at the molecular level to mechanistically delineating the distinct roles of SMAD1 and SMAD5 in directing cellular processes in the uterus.

Apart from directly regulating target gene expression, our data demonstrate that SMAD1/5 present as dense genomic occupancies. To date, only a limited amount of transcription factors have been investigated using the CUT&RUN-seq technique from the tissue samples due to antibody compatibility issue. We recognize that the binding sites and gene number identified here are quite high; however, the high density of binding events was also observed in the ENCODE (*Consortium E. P, 2012*) chromatin immunoprecipitation followed by sequencing (ChIP-seq) data for SMAD1 and SMAD5 in the human K562 cells, detecting an average of 63,563 peaks for SMAD1 and 109,682 peaks for SMAD5. (Data accessed through GSE95876 and GSE127365 from Gene Expression Omnibus.) Multiple aspects can contribute to the observation of dense SMAD1/5 genome occupancies. First, transcription factors (TFs) tend to dwell or 'search and bind' throughout the genome (*Chen et al., 2014*). Such events may not yield actual biological effects but rather are

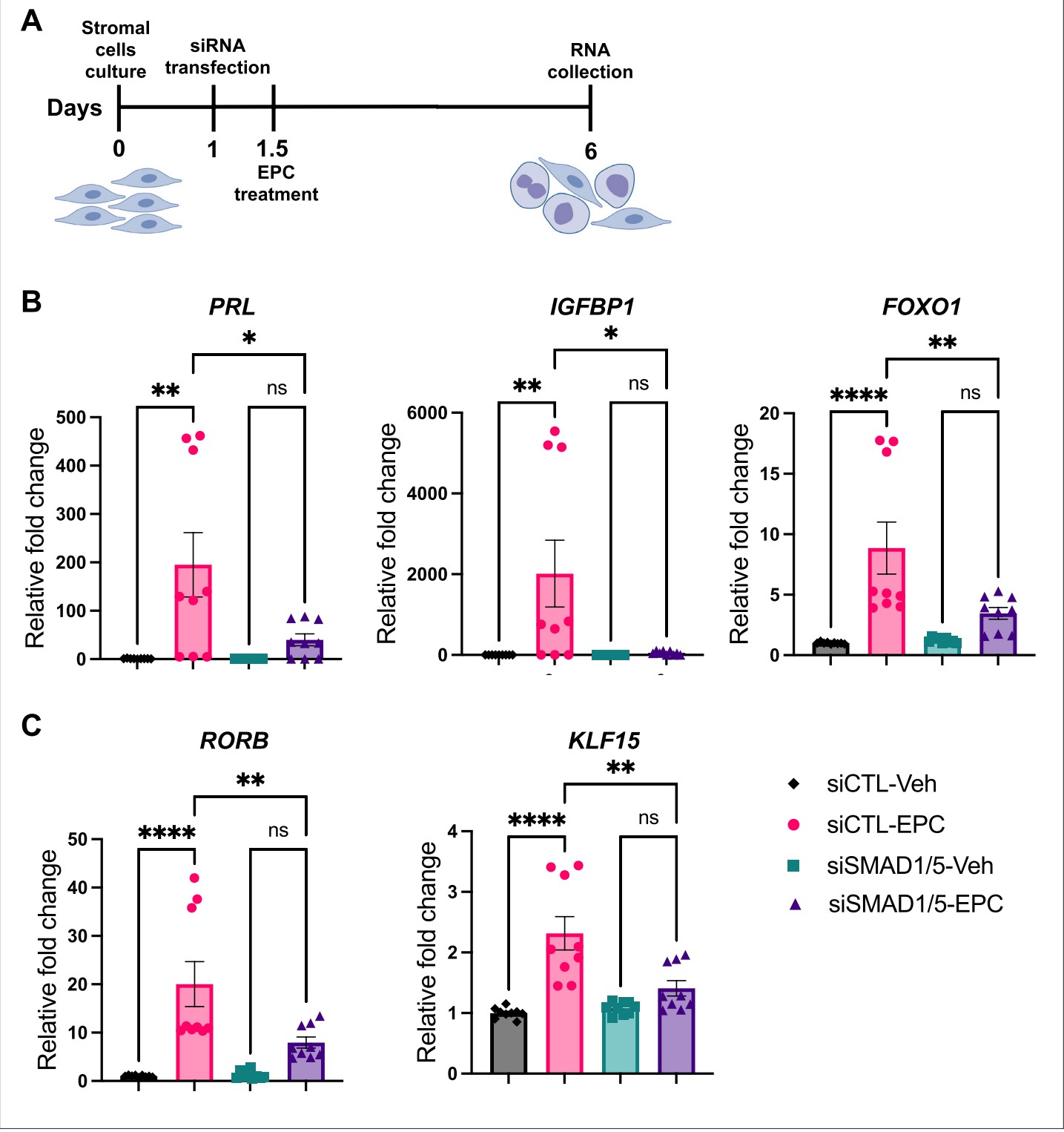

**Figure 6.** SMAD1 and SMAD5 are required for progesterone receptor (PR) responses during decidualization of human endometrial stromal cells (EnSCs). (**A**) Schematic approach and timeline outlining in vitro decidualization for EnSCs. (**B, C**) RT-qPCR results showing mRNA levels of *PRL*, *IGFBP1*, *FOXO1*, *RORB,* and *KLF15* after SMAD1/5 perturbation using siRNAs. Data are normalized to siCTL-Veh for visualization. Bar graphs represent average ± SEM of experiments on cells from three different individuals with technical triplicates. Analyzed by a one-way ANOVA with post hoc Tukey test.

The online version of this article includes the following figure supplement(s) for figure 6:

**Figure supplement 1.** Knockdown effect validation of SMAD1/5 perturbation.

due to differences in motif binding affinities (*Swinstead et al., 2016*). Second, apart from robust binding activities, TFs may not initiate transcription programs owing to the lack of co-factors or favorable conditions to exert their functions (*Chen et al., 2020*). Additionally, TF binding sites and target genes are unlikely to have a one-to-one relationship. TFs could be positioned from the proximal promoter regions to hundreds of kilobases afar to modulate gene expression. In the meantime, the same binding site could regulate multiple genes by interacting with different promoters in different subpopulations of cells. Lastly, TFs usually direct target gene expression in a cell-type-specific manner (*Arvey et al., 2012*). Our genomic profiling samples were collected from whole uterus at the time of 4.5 dpc, containing a great range of cell populations, including but not limited to the epithelium (luminal and glandular), stroma (progenitors and differentiated cells), myometrium, endothelium, and immune cell populations. The data is therefore expected to depict the dynamic and complex activities of SMAD1/5 in the entire uterus. Together, the stringent filtering and normalization criteria, comparable peak number to the published dataset, and IGV track view visualization collectively validate our CUT&RUN experiments and uncover the enriched regions as robust SMAD1/5 binding events. Our studies also examined the role of SMAD1/5 in mediating progesterone responses at the genomic and transcription levels. Similarly, our analysis was based on datasets generated from the whole mouse uterus, which contains multiple compartments of the uterine structures, including but not limited to epithelium and stroma. Published studies have shown that nuclear SMAD1/5 localize to the stroma and epithelium during the decidualization process at 4.5 dpc, during the window of implantation (*Monsivais et al., 2021*). Conditional deletion of SMAD1/5 exclusively in the uterine epithelium using lactoferrin-icre (*Ltf*-icre) results in severe subfertility due to impaired implantation and decidual development (*Tang et al., 2022*). Conditional deletion of SMAD1/5/4 exclusively in the cells from mesenchymal lineage (including uterine stroma) using anti-Mullerian hormone type 2 receptor cre (*Amhr2-cre*) results in infertility with defective decidualization (*Pangas et al., 2008*; *Rodriguez et al., 2016*). Given the essential roles of SMAD1/5 in both stroma and epithelium identified by previous studies, we believe that the transcriptional co-regulatory roles of SMAD1/5 and PR reported here using the whole uterus validate a relationship between SMAD1/5 and PR in both the stromal and epithelial compartments. However, it does not rule out potential co-regulatory roles of SMAD1/5 and PR in the myometrium, immune cells, and/or endothelium, given that whole uterus was used. The specific transcriptional evaluations of SMAD1/5 in the stroma versus the epithelium would require future validations using single-cell sequencing and/or spatial transcriptomic analysis.

Although our studies herein confirm that SMAD1 and SMAD5 proteins have distinct transcriptional regulatory activities, our previous studies demonstrated that while SMAD5 can functionally replace SMAD1, SMAD1 cannot replace SMAD5 in the uterus (*Monsivais et al., 2021*). How this epistatic relationship is established in a tissue-specific manner still needs to be determined by further biochemical investigations. In addition, further studies are needed to uncover whether SMAD1 and SMAD5 response differently upon ligand stimulation in the uterus, and if so, how the preference is achieved. Our study provides versatile in vivo genetic tools for these questions and can advance the toolbox for the field studying BMP signaling pathways. Because our mouse models are global knock-in mice, they will not only serve as a powerful tool for studying BMP signaling pathways in the reproductive system but will also promote the study of BMP signaling in other organs and tissues.

BMP signaling pathways are involved in a plethora of cellular processes and appropriate functioning of the BMP pathway depends on the precise crosstalk with other signaling pathways. Coordinated communication with other pathways can yield synergistic effects and lead to a complex regulatory network of biological processes. To be specific, SMAD1/5 mediates the crosstalk with the WNT/β-catenin pathway. WNT signaling inhibits glycogen synthase kinase 3β (GSK3β) activity and prevents SMAD1 from degradation, which governs the embryonic pattern formation (*Fuentealba et al., 2007*). Also, SMAD1/5 can physically interact with T-cell factor (TCF) or lymphoid enhancer factor (LEF) transcription factors to form transcriptional complexes to activate the transcription of many WNT-and BMP-responsive genes (*Labbé et al., 2000*). In addition, SMAD1 and SMAD5 can directly associate with Notch intracellular domain and enhance known Notch target gene expression by binding to their regulatory DNA sequences (*Zavadil et al., 2004*). Intriguingly, in prostate cells, SMAD1 physically interacts with the androgen receptor (AR) and halts the androgen-stimulated prostate cell growth (*Qiu et al., 2007*). Moreover, we provide first-hand evidence showing that BMP signaling pathways

converge with RA signaling pathways through the regulation of RORB by SMAD1/5. Further studies will grant a more detailed mechanism of the positive feedback loop between BMP and RA signaling.

Our previous studies suggest that the mouse endometrium presents decreased P4 responsiveness following the conditional deletion of SMAD1/5 in the uterus (*Monsivais et al., 2021*). In accordance with the phenotypical observation, we offer compelling support in our current study that SMAD1/5 work collectively with PR to regulate their target genes and that SMAD1/5 mediate the crosstalk between BMP and P4 signaling pathways during decidualization, a key process to ensure a successful pregnancy, and ultimately direct the biological transformations of the uterus during early pregnancy. We provide genomic evidence that SMAD1/5 are co-bound at around 35% of PR target genes in the mouse uterus during decidualization. Correspondingly, in a previously published study where they performed PR ChIP-seq in the mouse uterus after P4 stimulation, the SMAD1 motif was the fifth most significantly enriched sequence motifs identified (*Rubel et al., 2012*). In parallel, we also identified nuclear receptor motifs (i.e., PR sequence motifs) enriched in the SMAD1/5 binding sites (*Figure 5—figure supplement 1D and E*). From pathway enrichment analysis, we demonstrate that genes with SMAD1/5 and PR bound at the promoter regions are enriched for key pathways in directing the decidualization process, such as WNT and relaxin signaling pathways. Future studies can benefit from analyzing binding events beyond the promoter regions. Profiling the PR genome occupancy in the SMAD1/5-deficient mice would provide an interesting perspective to reevaluate the major regulatory roles of SMAD1/5 in mediating uterine transcriptomes. In this study, we determined the overlapped transcriptional control between SMAD1/5 and PR at the gene level, and functionally validated the regulatory effect at the transcript level in a human stromal cell decidualization model. While we observe a subset of peak representations that do not overlap at the base pair level in the promoter regions, future functional screenings at the promoter level, such as luciferase reporter assays to assess transcriptional co-activation by SMAD1/5 and PR, will advance this study.

SMADs are known to recruit co-repressors (i.e., Ski; *Luo et al., 1999*) or co-activators (i.e., p300; *Pouponnot et al., 1998*) to inhibit or activate target gene transcription, less is known about their cell-specific co-factors that confer the precise spatial-temporal control over binding activities to target genes. Our study highlights the potential co-factors by integrating both genomic and transcriptomic data to delineate signaling crosstalks that are responsible for maintaining tissue homeostasis, especially in the female reproductive tract.

Since mice only undergo decidualization upon embryo implantation whilst human stromal cells undergo cyclic decidualization in each menstrual cycle in response to rising levels of progesterone (*Ramathal et al., 2010*), asynchronous gene responses may occur in comparison between mouse models and human cells. However, cellular transformation during decidualization is conserved between mice and humans (*Gellersen and Brosens, 2014*), which makes findings in the mouse models a valuable and transferable resource to be evaluated in human tissues. Accordingly, our functional validation studies were performed using human EnSCs induced to decidualize in vitro for 4 d, which models the early phases of decidualization. Additional transcriptomic studies of the SMAD1/5 perturbations in human EnSCs will be of great resource in understanding the entire SMAD1/5 regulomes in humans.

In summary, our findings and those of others indicate that SMAD1 and SMAD5 not only are signal transducers for BMP signaling pathways, but also engage extensively in the crosstalk with PR signaling pathways. While P4 responses are critical for early pregnancy establishment, abnormal P4 responses are implicated in diseases such as endometriosis and endometrial cancers (*Brosens and Gellersen, 2006*; *Yilmaz and Bulun, 2019*; *Janzen et al., 2013*; *MacLean and Hayashi, 2022*). Hence, our results show that BMP and P4 signaling pathways synergize within the endometrium; these key pathways can shed light on the endometrial contribution to conditions that impact reproductive health in women, including early pregnancy loss, endometriosis, and endometrial cancer. Furthermore, we anticipate that the SMAD1/5 knock-in-tagged transgenic mouse models developed herein will be useful for studying BMP/SMAD1/5 signaling pathways in other reproductive and non-reproductive tract tissues in the body.

# Materials and methods

**Key resources table**

| Reagent type (species) or resource | Designation | Source or reference | Identifiers | Additional information |
|---|---|---|---|---|
| Strain, strain background (*Mus musculus*) | C57BL/6J × 129S5/SvEvBrd | This paper | C57BL/6J × 129S5/SvEvBrd | Can be obtained by contacting the corresponding authors |
| Transfected construct (*Homo sapiens*) | siCTL, si*SMAD1*, si*SMAD5* | Dharmacon | Cat# D-001810-10, L-012723-00-0005, L-015791-00-0005 | |
| Biological sample (*M. musculus*) | Primary uterine tissues | This paper | | Freshly isolated from 4.5 dpc mice. Can be obtained by contacting the corresponding authors |
| Antibody | Anti-HA (rabbit polyclonal) | EpiCypher | Cat# 13-2010, RRID:AB_3094663 | CUT&RUN (1:50) |
| Antibody | Anti-PA (rat monoclonal) | FUJIFILM Wako Pure Chemical Corporation | Cat# NZ-1, RRID:AB_3094664 | CUT&RUN (1:50) IP (10 ug/assay) WB (1:1000) |
| Antibody | Anti-HA (rabbit monoclonal) | Cell Signaling Technology | Cat# 3724, RRID:AB_1549585 | IP (1:50) WB (1:1000) |
| Antibody | Anti-SMAD1 (rabbit monoclonal) | Innovative Research | Cat# 385400, RRID:AB_431530 | WB (1:1000) |
| Antibody | Anti-SMAD5 (rabbit polyclonal) | ProteinTech | Cat# 12167-1-AP, RRID:AB_2286502 | WB (1:1000) |
| Sequence-based reagent | S1-F1 | MilliporeSigma | PCR primers | CAAACCGCAGACCAAGAAGC |
| Sequence-based reagent | S1-R1 | MilliporeSigma | PCR primers | CTTCTCCAGCTCTTCCATGGC |
| Sequence-based reagent | S5-F1 | MilliporeSigma | PCR primers | TGCTTAAGACCTGCATGTGACT |
| Sequence-based reagent | S5-R1 | MilliporeSigma | PCR primers | CATCCACTGCCTTTTCTGCC |
| Sequence-based reagent | *GAPDH*-F | MilliporeSigma | RT-qPCR primers | ACAACTTTGGTATCGTGGAAGG |
| Sequence-based reagent | *GAPDH*-R | MilliporeSigma | RT-qPCR primers | GCCATCACGCCACAGTTTC |
| Sequence-based reagent | *ACTB*-F | MilliporeSigma | RT-qPCR primers | CTGGAACGGTGAAGGTGACA |
| Sequence-based reagent | *ACTB*-R | MilliporeSigma | RT-qPCR primers | AAGGGACTTCCTGTAACAATGCA |
| Sequence-based reagent | *RPL13A*-F | MilliporeSigma | RT-qPCR primers | CCTGGAGGAGAAGAGGAAAGAGA |
| Sequence-based reagent | *RPL13A*-R | MilliporeSigma | RT-qPCR primers | TTGAGGACCTCTGTGTATTTGTCAA |
| Sequence-based reagent | *RORB*-F | MilliporeSigma | RT-qPCR primers | TGTGCCATCCAGATCACTCACG |
| Sequence-based reagent | *RORB*-R | MilliporeSigma | RT-qPCR primers | GGTTGAAGGCACGGCACATTCT |
| Sequence-based reagent | *SMAD5*-F | MilliporeSigma | RT-qPCR primers | CTCGCGAAAAGGAAGCTGTTG |
| Sequence-based reagent | *SMAD5*-R | MilliporeSigma | RT-qPCR primers | GGGTCAAGTCAGAGGCAGATT |

*Continued on next page*

*Continued*

| Reagent type (species) or resource | Designation | Source or reference | Identifiers | Additional information |
|---|---|---|---|---|
| Sequence-based reagent | *SMAD1*-F | MilliporeSigma | RT-qPCR primers | ATGGTGACACAGTTACTCGGT |
| Sequence-based reagent | *SMAD1*-R | MilliporeSigma | RT-qPCR primers | AGAGACTTCTTGGGTGGAAACA |
| Sequence-based reagent | *KLF15*-F | MilliporeSigma | RT-qPCR primers | GTGAGAAGCCCTTCGCCTGCA |
| Sequence-based reagent | *KLF15*-R | MilliporeSigma | RT-qPCR primers | ACAGGACACTGGTACGGCTTCA |
| Commercial assay or kit | PrimePCR SYBR Green Assay: *FOXO1*, Human | Bio-Rad | qHsaCED0004488 | |
| Commercial assay or kit | PrimePCR SYBR Green Assay: *IGFBP1*, Human | Bio-Rad | qHsaCID0014281 | |
| Commercial assay or kit | PrimePCR SYBR Green Assay: *PRL*, Human | Bio-Rad | qHsaCID0015557 | |

## Generation of knock-in mouse lines

*Smad5*$^{PA/PA}$ knock-in (KI) mice were generated using a similar approach as previously described (*Shimada et al., 2021*). Briefly, single-guide RNA (sg-RNA) was designed to target the regions close to the start codon (*Figure 1A and B*) and the sgRNA sequence was inserted into the pX459 V2.0 plasmid (#62988, Addgene). The reference plasmids containing PA tag sequence were constructed in pBluescript II SK (+) vector (Agilent, Palo Alto, CA). Then, 1 μg of guide RNA inserted vector and 1.0 μg of reference plasmid were co-transfected into EGRG01 embryonic stem (ES) cells. Out of 48 ES clones, 12 had the expected knock-in allele. ES cell clones that possessed the proper KI allele were injected into ICR embryos and chimeric blastocysts were transferred into pseudopregnant females. Chimeric male mice were mated with B6D2F1 female mice to obtain the PA-tagged SMAD5 KI heterozygous mice. Homozygous *Smad5*$^{PA/PA}$ mice were maintained in the C57BL/6J × 129S5/SvEvBrd mixed genetic background. To generate *Smad1*$^{HA/HA}$ mice, Cas9 protein (Thermo Fisher, A36497), sg-RNA, and a repair oligo of homology-directed repair (HDR) containing HA-tag and linker sequences were electroporated into zygotes harvested from in vitro fertilization using B6D2F1 male and female mice. An ECM830 electroporation system (BTX, Holliston, MA) was used for electroporation. Subsequently, embryos were cultured overnight to the two-cell stage and then transferred to the oviducts of pseudopregnant CD-1 mice (Center for Comparative Medicine, Baylor College of Medicine). Pups were further screened for successful heterozygous or homozygous knock-in alleles by PCR using primers spanning across the HA tag. Sequences of sgRNA, the single-stranded repair oligo for HDR, and primer used for genotype are listed in *Supplementary file 3a*.

## Animal ethics compliance and tissue collection

All mice were housed under standard conditions of a 12 hr light/dark cycle in a vivarium with controlled ambient temperature (70°F ± 2°F and 20–70% relative humidity). All mouse handling and experimental procedures were performed under AN-716 protocol approved by the Institutional Animal Care and Use Committee of Baylor College of Medicine. All experiments were performed with female mice aged between 7 and 12 wk with a C57BL/6J × 129S5/SvEvBrd mixed genetic background. All mice were euthanized using isoflurane induction followed by cervical dislocation, and tissues were snap-frozen in liquid nitrogen.

## CUT&RUN approach

Nuclei from uterine tissues were purified following a previously published protocol (*Fang et al., 2014*). The experiments were performed using pooled biological replicates from two mice that were processed as technical replicates throughout the CUT&RUN procedure and analysis. In short, uteri were harvested from pregnant mice at 4.5 dpc and washed with cold swelling buffer (10 mM Tris–HCl pH 7.5, 2 mM MgCl$_2$, 3 mM CaCl$_2$, 1X Protease Inhibitor Cocktail [PIC, Roche, 11836170001])

immediately after collection. Then tissue was cut into small pieces (~2–3 mm) using scissors, while submerged in cold swelling buffer. Nuclear extract was prepared by dounce homogenization in cold swelling buffer (using a size 7 dounce) and filtered using the cell strainer (100 μm, BD Biosciences). Lysate was centrifuged at 400 × g for 10 min, then resuspended in lysis buffer (swelling buffer with 10% glycerol and 1% CA-630, 1× PIC) using end-cut or wide-bore tips and incubated on ice for 5 min. Nuclei were washed twice with lysis buffer and resuspended in lysis buffer. Next, CUT&RUN procedure largely follows a previous protocol (*Skene and Henikoff, 2017*). Briefly, around 500,000 nuclei were used per reaction. 10 μl of concanavalin-coated beads (Bangs Labs, BP531) were washed twice in Bead Activation Buffer (20 mM HEPES, pH 7.9, 10 mM KCl, 1 mM CaCl$_2$, 1 mM MnCl$_2$) for each reaction. Then, beads were added to nuclei resuspension and incubated for 10 min at room temperature. After incubation, bead-nuclei complexes were resuspended in 100 μl Antibody Buffer (20 mM HEPES, pH 7.5, 150 mM NaCl, 0.5 mM spermidine, 1× PIC, 0.01% digitonin, and 2 mM EDTA) per reaction. Then, 1 μg of IgG antibody (Sigma, I5006), HA antibody (EpiCypher, 13-2010), PA antibody (Fuji Film, NZ-1), and PR antibody (Cell Signaling, D8Q2J) were added to each group respectively. After overnight incubation at 4°C, bead-nuclei complexes were washed twice with 200 μl cold Dig-Wash buffer (20 mM HEPES pH = 7.5, 150 mM NaCl, 0.5 mM spermidine, 1× PIC, 0.01% digitonin) and resuspended in 50 μl cold Dig-Wash buffer with 1 μl pAG-MNase (EpiCypher, 15-1016) per reaction. After incubation at room temperature for 10 min, bead-nuclei complexes were washed twice with 200 μl cold Dig-Wash buffer and resuspended in 50 μl cold Dig-Wash buffer, then 1 μl 100 mM CaCl$_2$ was added to each reaction. The mixture was incubated at 4°C for 2 hr and the reaction was stopped by adding 50 μl Stop Buffer (340 mM NaCl, 20 mM EDTA, 4 mM EGTA, 0.05% Digitonin, 100 ug/ml RNase A, 50 mg/ml glycogen, 0.5 ng *Escherichia coli* DNA Spike-in [EpiCypher, 18-1401]) and incubate at 37°C for 10 min. The supernatant was collected and subjected to DNA purification with phenol-chloroform and ethanol precipitation. Sequencing libraries were prepared using NEBNext Ultra II DNA Library Prep Kit (New England BioLabs, E7645) following the manufacturer's protocol. Paired-end 150 bp sequencing was performed on a NEXTSeq550 (Illumina) platform.

## Bioinformatic analysis for CUT&RUN data and reanalysis of published single-cell RNA-seq data

For CUT&Run data, raw data were de-multiplexed by bcl2fastq v2.20 with fastqc for quality control. Clean reads were mapped to reference genome mm10 by Bowtie2, with parameters of `--end-to-end --very-sensitive --no-mixed --no-discordant --phred33` -I 10 -X 700. For spike-in mapping, reads were mapped to *E. coli* genome U00096.3. Duplicated reads were removed, and only uniquely mapped reads were kept. Spike-in normalization was achieved through multiply primary genome coverage by scale factor (100,000/fragments mapped to *E. coli* genome). CUT&RUN peaks were called by SECAR (*Meers et al., 2019*) with the parameters of -norm -stringent -output. Track visualization was done by bedGraphToBigWig (*Kent et al., 2010*), bigwig files were imported to IGV for visualization. For peak annotation, common peaks were identified with 'mergePeaks' function in HOMER v4.11 (*Heinz et al., 2010*) and then genomic annotation was added by ChIPseeker (*Yu et al., 2015*). Motif analysis was conducted through HOMER v4.11 with parameter set as findMotifsGenome.pl mm10 -size 200 -mask (*Heinz et al., 2010*). Peak heatmaps were plotted using deepTools 2.4.2 with clustering options set to k-means (*Ramírez et al., 2016*). For imputing the cell fractions using published mouse uterine bulk and single-cell RNA-seq data, single-cell RNA-seq with cell type reference from 5.5 dpc mouse uterus was derived from GSE226417, bulk RNA-seq of the 3.5 dpc mouse uterus was derived from GSE152675. Signature matrix and cell fraction profiles were conducted through CIBERSORTx (*Newman et al., 2019*). Briefly, reference gene expression signature matrix was generated by 'Create Signature Matrix' module with default settings. Next, cell fractions were calculated using 'Impute Cell fractions' module in the absolute mode, which scales each annotated cellular fractions to an absolute value that reflects its absolute proportion in the bulk sample. For human single-cell RNA-seq data, raw data was obtained from EMBL-EBI under accession no. E-MTAB-10287. Cells with low coverage (less than 500 genes detected) were filtered, then gene counts were normalized for each cell by converting counts to quantiles and obtaining the corresponding values from a normal distribution. Then normalized cell vectors are concatenated along the gene panel. Plot visualization was conducted through CELLXGENE platform (*Abdulla et al., 2023*).

## Western blot analysis of immunoprecipitation (IP-WB)

Tissues were pulverized in liquid nitrogen and then lysed using NETN buffer (20 mM Tris–HCl, pH 8.0, 150 mM NaCl, 0.5 mM EDTA, 10% glycerol, and 0.5% NP-40). Protein concentration was determined by BCA Protein Assay Kit (Thermo Fisher, 23225). 1.5 mg of total protein lysate was used for IP. IP was performed by adding HA antibody (Cell Signaling, C29F4) or PA antibody (Fuji Film, NZ-1) to the lysate and incubate for 1 hr at 4°C. Subsequently, protein G magnetic beads (Thermo Fisher, 88847) were added for an additional 1 hr at 4°C. Then, the beads were washed five times with NETN buffer and denatured in sample buffer (Thermo Fisher, NP0007) for further analysis by western blot. For western blot procedures, briefly, denatured protein lysates were run on the 4 to 12%, Bis-Tris protein gels (Thermo Fisher, NP0321BOX) followed by electrophoretic transfer to nitrocellulose membrane. The membrane then went through blocking by 5% milk in Tris-buffered saline with Tween20 (TBST), followed by incubation overnight at 4°C in the primary antibodies anti-HA (Cell Signaling, C29F4), anti-PA (Fuji Film, NZ-1), anti-SMAD1 (Life Technologies, 385400), and anti-SMAD5 (ProteinTech, 12167-1-AP) at 1:1,000 dilution. The next day, membranes were washed three times with TBST, then incubated with horseradish peroxidase-conjugated secondary antibody for 1 hr at room temperature, then washed three times with TBST, developed and imaged on iBright Imaging System (FL1500).

## Primary endometrial stromal cells isolation/RNAi/decidualization

Studies using human specimens were conducted as indicated in a protocol approved by the Institutional Review Board at Baylor College of Medicine, H-51900. Human EnSCs were collected from healthy volunteers' menstrual effluent as previously reported (*Warren et al., 2018*; *Nayyar et al., 2020*; *Martínez-Aguilar et al., 2020*) (N = 3). In brief, samples were collected by participants in a DIVA cup during the 4–8 hr on the first night of menses and stored in DMEM/F12 with 10% FBS, antibiotic/antimycotic and 100 µg/ml Primocin in a cold-insulated pack until processing in the laboratory on the day of collection. The effluent was digested with 5 mg/ml collagenase and 0.2 mg/ml DNase I for 20 min at 37°C, then the cell pellet was collected by centrifuging at 2500 rpm for 5 min at room temperature. Next, red blood cell lysis was performed by resuspending the cell pellet in 20 ml of 0.2% NaCl for 20 s and neutralized with 20 ml of 1.6% NaCl. Then the solution was then centrifuged at 2500 rpm for 5 min. Then, 5 ml complete medium (DMEM/F12 supplement with 10% FBS, 1× Antibiotic-Antimycotic + 100 µg/ml Primocin) was used to resuspend the pellet and the solution was passed through 100 µm and 20 µm cell strainer sequentially. The flowthrough containing the stromal cells was centrifuged for 5 min at 2500 rpm and the pellet was resuspended in 10 ml complete medium and plated in a 10 cm dish. siRNA knockdown was performed using Lipofectamine RNAiMAX following the manufacturer's protocol. In brief, 0.2 million stromal cells were plated in a 12-well plate 1 d before transfection. On the day of transfection, 2 µl siRNA (20 µM, Dharmacon, D-001810-10, L-012723-00-0005, L-015791-00-0005) and 3 µl Lipofectamine RNAiMAX were diluted in 50 µl Opti-MEM respectively and then mixed to incubate at room temperature for 15 min. Then, the complex was added dropwise onto the cells. Also, 24 hr after transfection, medium was changed to DMEM/F12 supplement with 2% charcoal-stripped FBS. Decidualization was induced by the addition of 35 nM estradiol (Sigma, E1024), 1 µM medroxyprogesterone (Sigma, 1378001), and 0.05 mM cyclic adenosine monophosphate (Axxora, JBS-NU-1502L) for 4 d with media changes every 48 hr.

## RNA extraction and RT-qPCR

For mRNA extraction from stromal cells, cells were lysed with TRIzol and processed using the DirectZol kit (Zymo, R2051) following the manufacturer's procedures. Approximately 100 ng of mRNA was reverse transcribed into cDNA using iScript cDNA Supermix (Bio-Rad, 1708890) and amplified using specific primers listed in *Supplementary file 3a*. Primers were amplified using 2× SYBR Green Reagent (Life Technologies, 4364346) using a Bio-Rad CFX384 Touch Real-Time PCR Detection System. Data analysis was performed by calculating ΔΔCT value toward the geometric mean of *GAPDH, ACTB,* and *RPL13A* and then normalized to siCTL. p-Value was determined by one-way ANOVA with post hoc Tukey test using GraphPad Prism. *p≤0.05, **p≤0.01, ***p≤0.001, ****p≤0.0001.

## Acknowledgements

This research was supported by the Eunice Kennedy Shriver National Institute of Child Health and Human Development grants HD105800 (DM), HD096057 (DM), HD032067 (MMM), and HD110038 (MMM). DM is supported by a Next Gen Pregnancy Award from the Burroughs Wellcome Fund (NGP10125).

## Additional information

### Funding

| Funder | Grant reference number | Author |
|---|---|---|
| Eunice Kennedy Shriver National Institute of Child Health and Human Development | HD105800 | Diana Monsivais |
| Eunice Kennedy Shriver National Institute of Child Health and Human Development | HD096057 | Diana Monsivais |
| Eunice Kennedy Shriver National Institute of Child Health and Human Development | HD032067 | Martin Matzuk |
| Eunice Kennedy Shriver National Institute of Child Health and Human Development | HD110038 | Martin Matzuk |
| Burroughs Wellcome Fund | NGP10125 | Diana Monsivais |

The funders had no role in study design, data collection and interpretation, or the decision to submit the work for publication.

### Author contributions
Zian Liao, Conceptualization, Formal analysis, Validation, Methodology, Writing – original draft; Suni Tang, Data curation, Investigation, Visualization, Methodology, Writing – review and editing; Kaori Nozawa, Data curation, Investigation, Methodology, Writing – review and editing; Keisuke Shimada, Data curation, Methodology, Writing – review and editing; Masahito Ikawa, Resources, Project administration; Diana Monsivais, Conceptualization, Supervision, Funding acquisition, Investigation, Writing – review and editing; Martin Matzuk, Conceptualization, Supervision, Funding acquisition, Writing – review and editing

### Author ORCIDs
Zian Liao ⓘ https://orcid.org/0000-0002-7198-2182
Keisuke Shimada ⓘ https://orcid.org/0000-0003-3739-7163
Masahito Ikawa ⓘ http://orcid.org/0000-0001-9859-6217
Diana Monsivais ⓘ http://orcid.org/0000-0001-5660-6392
Martin Matzuk ⓘ https://orcid.org/0000-0002-1445-8632

### Ethics
All mouse handling and experimental procedures were performed under the AN-716 protocol approved by the Institutional Animal Care and Use Committee of Baylor College of Medicine. Studies using human specimens were conducted as indicated in a protocol approved by the Institutional Review Board at Baylor College of Medicine, H-51900.

Reviewer #2 (Public Review): https://doi.org/10.7554/eLife.91434.4.sa1
Author Response https://doi.org/10.7554/eLife.91434.4.sa2

# Additional files

## Supplementary files

• Supplementary file 1. Complete list of motif enrichment analysis from the up-targets and down-targets for SMAD1.

• Supplementary file 2. Complete list of motif enrichment analysis from the up-targets and down-targets for SMAD5.

• Supplementary file 3. Tables with detailed information mentioned in the main text. **(a)** DNA sequences used in the study. **(b)** Direct target genes of SMAD1/5. **(c)** Results of Gene Ontology enrichment analysis of shared direct target genes of SMAD1/5 from the up-targets. **(d)** Results of Gene Ontology enrichment analysis of shared direct target genes of SMAD1/5 from the down-targets. **(e)** Results of KEGG pathway enrichment analysis for shared genes bound by SMAD1, SMAD5, and PR. **(f)** Results of motif analysis results for shared peaks between SMAD1, SMAD5, and PR. **(g)** Gene overlapping results of SMAD1/5 and SOX17. **(h)** Gene overlapping results of SMAD1/5 and NR2F2.

• MDAR checklist

## Data availability

Sequencing data and analyses are deposited in the Gene Expression Omnibus under accession number GSE237975.

The following dataset was generated:

| Author(s) | Year | Dataset title | Dataset URL | Database and Identifier |
|---|---|---|---|---|
| Liao Z, Tang S, Nozawa K, Shimada K, Ikawa M, Monsivais D, Matzuk MM | 2024 | Utilization of Tagged Transgenic Mouse Lines to Study the Molecular Roles of SMAD1/5 in Mediating Signaling Crosstalk During Early Pregnancy II | https://www.ncbi.nlm.nih.gov/geo/query/acc.cgi?acc=GSE237975 | NCBI Gene Expression Omnibus, GSE237975 |

The following previously published datasets were used:

| Author(s) | Year | Dataset title | Dataset URL | Database and Identifier |
|---|---|---|---|---|
| Yang M, Ong J | 2023 | Spatiotemporal insight into early pregnancy governed by immune-featured stromal cells [scRNA-seq] | https://www.ncbi.nlm.nih.gov/geo/query/acc.cgi?acc=GSE226417 | NCBI Gene Expression Omnibus, 226417 |
| Creighton C, Chen F, Monsivais D, Matzuk M | 2021 | Gene expression profiling of Smad1/5 cKO and Acvr2a cKO mice | https://www.ncbi.nlm.nih.gov/geo/query/acc.cgi?acc=GSE152675 | NCBI Gene Expression Omnibus, 152675 |

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
