## [Editor Report · eLife assessment]

This study presents two **valuable** new mouse models that individually tag proteins from the SMAD family to identify distinct roles during early pregnancy. **Convincing** evidence is provided that SMAD1 and SMAD5 target many of the same genomic regions as each other and the progesterone receptor. Given the broad effect of these signaling pathways in multiple systems, these new tools will most likely interest readers across biological disciplines.

---

## [Referee Report · Reviewer #2 (Public Review)]

Summary:

Liao and colleagues generated tagged SMAD1 and SMAD5 mouse models and identified genome occupancy of these two factors in the uterus of these mice using the CUT&RUN assay. The authors used integrative bioinformatic approaches to identify putative SMAD1/5 direct downstream target genes and to catalog the SMAD1/5 and PGR genome co-localization pattern. The role of SMAD1/5 on stromal decidualization was assayed in vitro on primary human endometrial stromal cells. The new mouse models offer opportunities to further dissect SMAD1 and SMAD5 functions without the limitation from SMAD antibodies, which is significant. The CUT&RUN data further support the usefulness of these mouse models for this purpose.

Strengths:

The strength of this study is the novelty of new mouse models and the valuable cistromic data derived from these mice. This revised manuscript provides lots of food for thought inside and outside of the field of reproductive biology.

Weaknesses:

Causal effects of SMAD1/5 on the genome occupancy of other major uterine transcription factors were discussed but not experimentally examined in the present manuscript, which is understandable.

---

## [Author Response]

The following is the authors’ response to the previous reviews.

**Public Reviews:**

**Reviewer #1 (Public Review):**
SummaryLiao et al leveraged two powerful genomics techniques-CUT&RUN and RNA sequencing-to identify genomic regions bound by and activated or inactivated by SMAD1, SMAD5, and the progesterone receptor during endometrial stromal cell decidualization. Additionally, the authors generated novel knock-in HA-SMAD1 and PA-SMAD5 tagged mice to combat antibody issues facing the field, generating a novel model to advance the study of BMP signaling in the female reproductive tract. During decidualization in a murine model, SMAD1/5 are bound to many genomic sites of genes important in decidualization and pregnancy and coregulated responses with progesterone receptor signaling.StrengthsThe authors utilized powerful next generation sequencing and identified important transcriptional mechanisms of SMAD1/5 and PGR during decidualization in vivo.WeaknessesNone.Overall, the manuscript and study are well structured and provide critical mechanistic updates on the roles of SMAD1/5 in decidualization and preparation of the maternal endometrium for pregnancy.

We thank you for the summary and consideration.

**Reviewer #2 (Public Review):**
Summary:Liao and colleagues generated tagged SMAD1 and SMAD5 mouse models and identified genome occupancy of these two factors in the uterus of these mice using the CUT&RUN assay. The authors used integrative bioinformatic approaches to identify putative SMAD1/5 direct downstream target genes and to catalog the SMAD1/5 and PGR genome co-localization pattern. The role of SMAD1/5 on stromal decidualization was assayed in vitro on primary human endometrial stromal cells. The new mouse models offer opportunities to further dissect SMAD1 and SMAD5 functions without the limitation from SMAD antibodies, which is significant. The CUT&RUN data further support the usefulness of these mouse models for this purpose.Strengths:The strength of this study is the novelty of new mouse models and the valuable cistromic data derived from these mice. Overall the present manuscript is an excellent resource paper for the field of reproductive biology.Weaknesses:The weakness of the present version of the manuscript includes the self-limited data analysis approaches such as the proximal promoter based bioinformatic filter and an outdated method on inferring the cell type composition. Evidence was provided for potential associations between SMAD1/5 and other major transcription factors. However, causal effects of SMAD1/5 on the genome occupancy of other major uterine transcription factors were discussed but not experimentally examined in the present manuscript, which is understandable.For data in Figure 2B, the current manuscript fails to elaborate the common and distinct features between clusters 1 and 3 as well as the biological significance of having two separate clusters for SMAD1. In addition, Figure S1A shows overlapping genome occupancy between SMAD1 and SMAD5, which is not clearly demonstrated in Figure 2B.

Thank you for the comments. We’ve added additional interpretations in Lines 281-283, addressing the clustering results mentioned in Figure 2B as suggested. We do appreciate the overlapping genome occupancy in Cluster 1, although the signal intensities may differ between two groups.

Lines 281-283:

“Peaks in cluster 1 exhibit a shared enrichment for both SMAD1 and SMAD5, whereas clusters 2 and 3 demonstrate preferential enrichment for SMAD5 and SMAD1, respectively.”

For data in Figure 5A, the result description does not provide adequate information to guide readers to full understanding of the data. The biological meaning behind the three PR clusters is not stated nor speculated. Moreover, Figure 5A and Figure S1B are inherently connected but fail to be adequately described in the main text.

Thank you for the comments. We’ve added additional interpretations in Lines 415-421 discussing the clustering results mentioned in Figure 5A, together with Supplement Figure 1C (Former Supplement Figure 1B) as suggested.

Lines 415-421:

“Based on the k-means clustering results of the peaks, we demonstrated clusters with shared occupancy between SMAD1/5 and PR (cluster 1), preferential deposition in the SMAD1 (cluster2), SMAD5 (cluster 4) and PR (clusters 3,5), respectively. Interestingly, between clusters 3 and 5, although the primary enrichment is for PR, overall, the signal intensities for SMAD5 are higher in cluster 5. Together with previous analysis on genes uniquely or commonly bound by SMAD1/5 (Supplement Figure 1A), we speculate such observation can be attributed to a subset of the genes that are potentially co-regulated by SMAD5 and PR.”

**Reviewer #3 (Public Review):**
Summary:As SMAD1/5 activities have previously been indistinguishable, these studies provide a new mouse model to finally understand unique downstream activation of SMAD1/5 target genes, a model useful for many scientific fields. Using CUT&RUN analyses with gene overlap comparisons and signaling pathway analyses, specific targets for SMAD1 versus SMAD5 were compared, identified, and interpreted. These data validate previous findings showing strong evidence that SMADs directly govern critical genes required for endometrial receptivity and decidualization, including cell adhesion and vascular development. Further, SMAD targets were overlapped with progesterone receptor binding sites to identify regions of potential synergistic regulation of implantation. The authors report strong correlations between progesterone receptor and SMAD1/5 direct targets to cooperatively promote embryo implantation. Finally, the authors validated SMAD1/5 gene regulation in primary human endometrial stromal cells. These studies provide a data-rich survey of SMAD family transcription, defining its role as a governor of early pregnancy.Strengths:This manuscript provides a valuable survey of SMAD1/5 direct transcriptional events at the time of receptivity. As embryo implantation is controlled by extensive epithelial to stromal molecular crosstalk and hormonal regulation in space and time, the authors state a strong, descriptive narrative defining how SMAD1/5 plays a central role at the site of this molecular orchestration. The implementation of cutting-edge techniques and models and simple comparative analyses provide a straightforward, yet elegant manuscript.Although the progesterone receptor exists as a major regulator of early pregnancy, the authors have demonstrated clear evidence that progesterone receptor with SMAD1/5 work in concert to molecularly regulate targets such as Sox17, Id2, Tgfbr2, Runx1, Foxo1 and more at embryo implantation. Additionally, the authors pinpoint other critical transcription factor motifs that work with SMADs and the progesterone receptor to promote early pregnancy transcriptional paradigms.Weaknesses:Although a wonderful new tool to ascertain SMAD1 versus SMAD5 downstream signaling, the importance of these factors in governing early pregnancy is not novel. Furthermore, functional validation studies are needed to confirm interactions at promoter regions. Additionally, the authors presume that all overlapped genes are shared between progesterone receptor and SMAD1/5, yet some peak representations do not overlap. Although, transcriptional activation can occur at the same time, they may not occur in the same complex. Thus, further confirmation of these transcriptional events is warranted.

Thank you for the comments. We recognized this limitation and discussed future options regarding this in Lines 578-583.

Lines 578-583:

“In this study, we determined the overlapped transcriptional control between SMAD1/5 and PR at the gene level, and functionally validated the regulatory effect at the transcript level in a human stromal cell decidualization model. While we observe a subset of peak representations that do not overlap at the base pair level in the promoter regions, future functional screenings at the promoter level, such as luciferase reporter assays to assess transcriptional co-activation by SMAD1/5 and PR, will advance this study.”

Since whole murine uterus was used for these studies, the specific functions of SMAD1/5 in the stroma versus the epithelium (versus the myometrium) remain unknown. Further work is needed to delineate binding and transcriptional activation of SMAD1/5 and the progesterone receptor in the uterine compartments.

We thank the reviewer for the insightful comment. Given the multifaceted roles of SMAD1/5 play the female reproductive tract, we concur that future studies will benefit from a more compartmentalized approach, as discussed in Lines 526-538.

Lines 526-538:

“Published studies have shown that nuclear SMAD1/5 localize to the stroma and epithelium during the decidualization process at 4.5 dpc, during the window of implantation. Conditional deletion of SMAD1/5 exclusively in the uterine epithelium using lactoferrin-icre (Ltf-icre) results in severe subfertility due to impaired implantation and decidual development. Conditional deletion of SMAD1/5/4 exclusively in the cells from mesenchymal lineage (including uterine stroma) using anti-Mullerian hormone type 2 receptor cre (Amhr2-cre) results in infertility with defective decidualization. Given the essential roles of SMAD1/5 in both stroma and epithelium identified by previous studies, we believe that the transcriptional co-regulatory roles of SMAD1/5 and PR reported here using the whole uterus validates a relationship between SMAD1/5 and PR in both the stromal and epithelial compartments. However, it does not rule out potential coregulatory roles of SMAD1/5 and PR in the myometrium, immune cells, and/or endothelium, given that whole uterus was used. The specific transcriptional evaluations of SMAD1/5 in the stroma versus the epithelium would require future validations using single-cell sequencing and/or spatial transcriptomic analysis.”

There are asynchronous gene responses in the SMAD1/5 ablated mouse model compared to the siRNA-treated human endometrial stromal cells. These differences can be confounding. Further investigation is needed to understand the meaning of these differences and as they relate to the entire SMAD transcriptome.

Thank you for the comments. In the current study, we used human endometrial stromal cells as a model to validate our findings functionally, aiming to mimic the specific time point during decidualization. We acknowledge the similarities and differences between the mouse and human cell models, and this information needs to be considered when evaluating genome-wide effects on the transcriptome. This point is discussed ins Lines 589-597.

Lines 589-597:

“Since mice only undergo decidualization upon embryo implantation whilst human stromal cells undergo cyclic decidualization in each menstrual cycle in response to rising levels of progesterone, asynchronous gene responses may occur in comparison between mouse models and human cells. However, cellular transformation during decidualization is conserved between mice and humans, which makes findings in the mouse models a valuable and transferable resource to be evaluated in human tissues. Accordingly, our functional validation studies were performed using human endometrial stromal cells induced to decidualize in vitro for four days, which models the early phases of decidualization. Additional transcriptomic studies of the SMAD1/5 perturbations in human endometrial stromal cells will be of great resource in understanding the entire SMAD1/5 regulomes in humans.”

**Recommendations for the authors:**

**Reviewer #2 (Recommendations For The Authors):**
The inference on the cell type composition could use updated bioinformatic tools, which are purely computational without costly and time-consuming wet-lab resources. Perhaps this part of the description could be streamlined if the authors chose to use the method in the current version.

We thank the reviewer for the suggestion. We added the analysis of the cell type composition using the updated tool CIBERSORTx (PMID:31061481) and included the results and discussion regarding the cell type composition changes in Supplement Figure 1B and Lines 392-407.

Lines 392-407

“To explore the major cell types regulated by SMAD1/5, first, we used CIBERSORTx to analyze and depict changes in the cell populations upon SMAD1/5 depletion in the mouse uterus during early pregnancy. By imputing the bulk uterine gene expression profiles to previously published mouse uterine single-cell datasets using CIBERSORTx, we were able to compare changes across both samples and cell types upon the SMAD1/5 perturbation in the mouse uterus. We highlight the proportional increase in the epithelial cells, as well as the decrease in the decidual stromal cells and smooth muscle cells in mice lacking uterine SMAD1/5 during the periimplantation phase (Supplement Figure 1B). Such cell populational changes are in line with the phenotypical observations of decidualization failure and excessive proliferation in the epithelial compartment. In addition, to explore the expression patterns of SMAD1/5 direct targets in human, we profiled the expression levels of the key “up-targets” and “down-targets” in the different cell types of the human endometrium. Using previously published single-cell RNA seq data of human endometrium, we visualized the expression patterns of suppressive targets and activating targets of SMAD1/5 (Figure 4E). Apart from the major epithelial and stromal compartments, SMAD1/5 target genes are also widely expressed in the immune cell populations. Such observations reinforced the importance of the BMP signaling pathways in establishing an immune-privileged environment at the maternal-fetal interface.”